# Robust Density Estimation under Besov IPM Losses

**Ananya Uppal**
Department of Mathematical Sciences
Carnegie Mellon University
auppal@andrew.cmu.edu

**Shashank Singh**
Machine Learning Department
Carnegie Mellon University
shashanksi@google.com

**Barnabás Póczos**
Machine Learning Department
Carnegie Mellon University
bapoczos@cs.cmu.edu

## Abstract

We study minimax convergence rates of nonparametric density estimation under the Huber contamination model, in which a proportion of the data comes from an unknown outlier distribution. We provide the first results for this problem under a large family of losses, called Besov integral probability metrics (IPMs), that include the $\mathcal{L}^p$, Wasserstein, Kolmogorov-Smirnov, Cramer-von Mises, and other commonly used metrics. Under a range of smoothness assumptions on the population and outlier distributions, we show that a re-scaled thresholding wavelet estimator converges at minimax optimal rates under a wide variety of losses and also exhibits optimal dependence on the contamination proportion. We also provide a purely data-dependent extension of the estimator that adapts to both an unknown contamination proportion and the unknown smoothness of the true density. Finally, based on connections recently shown between density estimation under IPM losses and generative adversarial networks (GANs), we show that certain GAN architectures are robustly minimax optimal.

## 1 Introduction

In many settings, observed data contains not only samples from the distribution of interest, but also a small proportion of outlier samples. Because these outliers can exhibit arbitrary, unpredictable behavior, they can be difficult to detect or to explicitly account for. This has inspired a large body of work on *robust statistics*, which seeks statistical methods for which the error introduced by a small proportion of arbitrary outlier samples can be controlled.

The majority of work in robust statistics has focused on providing guarantees under the Huber $\epsilon$-contamination model [Huber, 1965]. Under this model, data is assumed to be observed from a mixture distribution $(1 - \epsilon)P + \epsilon G$, where $P$ is an unknown population distribution of interest, $G$ is an unknown outlier distribution, and $\epsilon \in [0, 1)$ is the "contamination proportion" of outlier samples. Equivalently, this models the misspecified case in which data are drawn from a small perturbation by $\epsilon(G - P)$ of the target distribution $P$ of interest. The goal is then to develop methods whose performance degrades as little as possible when $\epsilon$ is non-negligible.

The present paper studies nonparametric density estimation under this model. Specifically, given independent and identically distributed samples from the mixture $(1 - \epsilon)P + \epsilon G$, we characterize minimax optimal convergence rates for estimating $P$. Prior work on this problem has assumed $P$ has a Hölder continuous density $p$ and has provided minimax rates under total variation loss [Chen et al., 2018] or for estimating $p(x)$ at a point $x$ [Liu and Gao, 2017]. In the present paper, in addition

to considering a much wider range of smoothness conditions (characterized by $p$ lying in a Besov space), we provide results under a large family of losses called integral probability metrics (IPMs);

$$d_{\mathcal{F}}(P, Q) = \sup_{f \in \mathcal{F}} \left| \underset{X \sim P}{\mathbb{E}} f(X) - \underset{X \sim Q}{\mathbb{E}} f(X) \right|, \tag{1}$$

where $P$ and $Q$ are probability distributions and $\mathcal{F}$ is a "discriminator class" of bounded Borel functions. As shown in several recent papers [Liu et al., 2017, Liang, 2018, Singh et al., 2018, Uppal et al., 2019], IPMs play a central role not only in nonparametric statistical theory and empirical process theory, but also in the theory of generative adversarial networks (GANs). Hence, this work advances not only basic statistical theory but also our understanding of the robustness properties of GANs.

In this paper, we specifically discuss the case of Besov IPMs, in which $\mathcal{F}$ is a Besov space (see Section 2.1). In classical statistical problems, Besov IPMs provide a unified formulation of a wide variety of distances, including $\mathcal{L}^p$ [Wasserman, 2006, Tsybakov, 2009], Sobolev [Mroueh et al., 2017, Leoni, 2017], maximum mean discrepancy (MMD; [Tolstikhin et al., 2017])/energy [Székely et al., 2007, Ramdas et al., 2017], Wasserstein/Kantorovich-Rubinstein [Kantorovich and Rubinstein, 1958, Villani, 2008], Kolmogorov-Smirnov [Kolmogorov, 1933, Smirnov, 1948], and Dudley metrics [Dudley, 1972, Abbasnejad et al., 2018]. Hence, as we detail in Section 4.3, our bounds for robust nonparametric density estimation apply under many of these losses. More recently, it has been shown that generative adversarial networks (GANs) can be cast in terms of IPMs, such that convergence rates for density estimation under IPM losses imply convergence rates for certain GAN architectures [Liang, 2018, Singh et al., 2018, Uppal et al., 2019]. Thus, as we show in Section 5, our results imply the first robustness results for GANs in the Huber model.

In addition to showing rates in the classical Huber model, which avoids assumptions on the outlier distribution $G$, we consider how rates change under additional assumptions on $G$. Specifically, we show faster convergence rates are possible under the assumption that $G$ has a bounded density $g$, but that these rates are not further improved by additional smoothness assumptions on $g$.

Finally, we overcome a technical limitation of recent work studying density estimation under Besov IPMs losses. Namely, the estimators used in past work rely on the unrealistic assumption that the practitioner knows the Besov space in which the true density lies. This paper provides the first convergence rates for a purely data-dependent density estimator under Besov IPMs, as well as the first nonparametric convergence guarantees for a fully data-dependent GAN architecture.

## 1.1 Paper Organization

The rest of this paper is organized as follows. Section 2 formally states the problem we study and defines essential notation. Section 3 discusses related work in nonparametric density estimation. Section 4.1 contains minimax rates under the classical "unstructured" Huber contamination model, while Section 4.2 studies how these rates change when additional assumptions are made on the contamination distribution. Section 4.3 develops our general results from Sections 4.1 and 4.2 into concrete minimax convergence rates for important special cases. Finally, Section 5 applies our theoretical results to bound the error of perfectly optimized GANs in the presence of contaminated data. All theoretical results are proven in the Appendix.

## 2 Formal Problem Statement

We now formally state the problems studied in this paper. Let $p$ be a density of interest and $g$ be the contamination density such that $X_1, \ldots, X_n \sim (1 - \epsilon)p + \epsilon g$ are $n$ IID samples. We wish to use these samples to estimate $p$. We consider two qualitatively different types of contamination, as follows.

In the "unstructured" or Huber contamination setting, we assume that $p$ lies in some regularity class $\mathcal{F}_g$, but $g$ may be any compactly supported density. In particular, we assume that the data is generated from a density living in the set $\mathcal{M}(\epsilon, \mathcal{F}_g) = \{(1 - \epsilon)p + \epsilon g : p \in \mathcal{F}_g, g \text{ has compact support}\}$. We then wish to bound the minimax risk of estimating $p$ under an IPM loss $d_{\mathcal{F}_d}$; i.e., the quantity

$$\mathcal{R}(n, \epsilon, \mathcal{F}_g, \mathcal{F}_d) = \inf_{\widehat{p}_n} \sup_{f \in \mathcal{M}(\epsilon, \mathcal{F}_g)} \underset{f}{\mathbb{E}} [d_{\mathcal{F}_d}(p, \widehat{p}_n)] \tag{2}$$

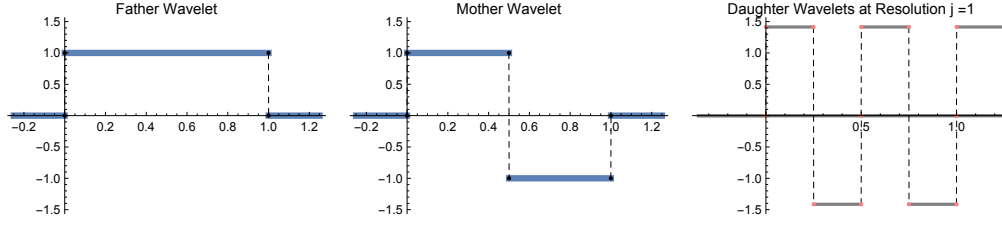

Figure 1: Father, Mother, and first few Daughter elements of the Haar Wavelet Basis.

where the infimum is taken over all estimators $\widehat{p}_n$.

In the "structured" contamination setting, we additionally assume that the contamination density $g$ lives in a smoothness class $\mathcal{F}_c$. The data is generated by a density in $\mathcal{M}(\epsilon, \mathcal{F}_g, \mathcal{F}_c) = \{(1-\epsilon)p + \epsilon g : p \in \mathcal{F}_g, g \in \mathcal{F}_c\}$ and we seek to bound the minimax risk

$$\mathcal{R}(n, \epsilon, \mathcal{F}_g, \mathcal{F}_c, \mathcal{F}_d) = \inf_{\widehat{p}_n} \sup_{f \in \mathcal{M}(\epsilon, \mathcal{F}_g, \mathcal{F}_c)} \mathbb{E}_f \left[ d_{\mathcal{F}_d}(\widehat{p}_n, p) \right]. \tag{3}$$

In the following section, we provide notation to formalize the spaces $\mathcal{F}_g, \mathcal{F}_c$ and $\mathcal{F}_d$ that we consider.

## 2.1 Set up and Notation

For non-negative real sequences $\{a_n\}_{n\in\mathbb{N}}$, $\{b_n\}_{n\in\mathbb{N}}$, $a_n \lesssim b_n$ indicates $\limsup_{n\to\infty} \frac{a_n}{b_n} < \infty$, and $a_n \asymp b_n$ indicates $a_n \lesssim b_n \lesssim a_n$. For $q \in [1, \infty]$, $q' := \frac{q}{q-1}$ denotes the Hölder conjugate of $q$ (with $1' = \infty$, $\infty' = 1$). $\mathcal{L}^q(\mathbb{R}^D)$ (resp. $l^q$) denotes the set of functions $f$ (resp. sequences $a$) with $\|f\|_q := \left( \int |f(x)|^q \, dx \right)^{1/q} < \infty$ (resp. $\|a\|_{l^q} := \left( \sum_{n\in\mathbb{N}} |a_n|^q \right)^{1/q} < \infty$).

We now define the family of Besov spaces studied in this paper. Besov spaces generalize Hölder and Sobolev spaces and are defined using wavelet bases. As opposed to the Fourier basis, wavelet bases provide a localization in space as well as frequency which helps express spatially inhomogeneous smoothness.

A wavelet basis is formally defined by the mother $(\psi(x))$ and father$(\phi(x))$ wavelets. The basis consists of two parts; first, the set of translations of the father and mother wavelets i.e.

$$\Phi = \{\phi(x - k) : k \in \mathbb{Z}^d\} \tag{4}$$

$$\Psi = \{\psi_\epsilon(x - k) : k \in \mathbb{Z}^d, \epsilon \in \{0, 1\}^D\}, \tag{5}$$

second, the set of daughter wavelets, i.e.,

$$\Psi_j = \{2^{Dj/2}\psi_\epsilon(2^{Dj}x - k) : k \in \mathbb{Z}^d, \epsilon \in \{0, 1\}^D\}. \tag{6}$$

Then the union $\Phi \cup \Psi \cup (\bigcup_{j\geq 0})\Psi_j$ is an orthonormal basis for $\mathcal{L}^2(\mathbb{R}^D)$.

We defer the technical assumptions on the mother and father wavelet to the appendix. Instead for intuition, we illustrate in Figure 1 the few terms of the best-known wavelet basis of $\mathcal{L}^2(\mathbb{R})$, the Haar wavelet basis.

In higher dimensions, the wavelet basis is defined using the tensor product of wavelets in dimension 1. For details, see, Härdle et al. [2012] and Meyer [1992].

To effectively express smooth functions we will require $r$-regular ($r$-regularity is precisely defined in the appendix) wavelets. We assume throughout our work that $\phi$ and $\psi$ are compactly supported $r$-regular wavelets. We now formally define a Besov space.

**Definition 1** (Besov Space). *Given an $r$-regular wavelet basis of $\mathcal{L}^2(\mathbb{R}^D)$, let $0 \leq \sigma < r$, and $p, q \in [1, \infty]$. Then the Besov space $B_{p,q}^\sigma(\mathbb{R}^D)$ is defined as the set of functions $f : \mathbb{R}^D \to \mathbb{R}$ that satisfy*

$$\|f\|_{B_{p,q}^\sigma} := \|\alpha\|_{l^p} + \left\| \left\{ 2^{j(\sigma + D(1/2 - 1/p))} \|\beta_j\|_{l^p} \right\}_{j\in\mathbb{N}} \right\|_{l^q} < \infty \tag{7}$$

*where $\alpha$ is the set of vectors $\{\alpha_\phi\}_{\phi\in\Phi}$ where $\alpha_\phi := \int_{\mathbb{R}^D} f(x)\phi(x)dx$ and $\beta_j$ is the set of vectors $\{\beta_\psi\}_{\psi\in\Psi_j}$, where $\beta_\psi := \int_{\mathbb{R}^D} f(x)\psi(x)dx$.*

The quantity $\|f\|_{B_{p,q}^\sigma}$ is called the *Besov norm of $f$*. For any $L > 0$, we write $B_{p,q}^\sigma(L)$ to denote the closed Besov ball $B_{p,q}^\sigma(L) = \{f \in B_{p,q}^\sigma : \|f\|_{B_{p,q}^\sigma} \le L\}$. When the constant $L$ is unimportant (e.g., for *rates* of convergence), $B_{p,q}^\sigma$ denotes a ball $B_{p,q}^\sigma(L)$ of finite but arbitrary radius $L$. We provide well-known examples from the rich class of resulting spaces in Section 4.3.

We now define "linear (distribution) estimators", a commonly used sub-class of distribution estimators:

**Definition 2** (Linear Estimator). *Let $(\Omega, \mathcal{F}, P)$ be a probability space. An estimate $\widehat{P}$ of $P$ is said to be* linear *if there exist functions $T_i(X_i, \cdot) : \mathcal{F} \to \mathbb{R}$ such that for all measurable $A \in \mathcal{F}$, $\widehat{P}(A) = \sum_{i=1}^n T_i(X_i, A)$.*

Common examples of linear estimators are the empirical distribution, the kernel density estimator and the linear wavelet series estimator considered in this paper.

## 3 Related Work

This paper extends recent results in both non-parametric density estimation and robust estimation. We now summarize the results of the most relevant papers, namely those of Uppal et al. [2019], Chen et al. [2016], and Liu and Gao [2017].

### 3.1 Nonparametric Density Estimation under Besov IPM Losses

Uppal et al. [2019] studied the estimation of a density lying in a Besov space $B_{p_g, q_g}^{\sigma_g}$ under Besov IPM loss $d_{B_{p_d, q_d}^{\sigma_d}}$ with uncontaminated data. As shorthand, we will write $\mathcal{R}(n, \mathcal{F}_g, \mathcal{F}_d) = \mathcal{R}(n, 0, \mathcal{F}_g, \mathcal{F}_d)$ and $\mathcal{R}(n, \mathcal{F}_g, \mathcal{F}_c, \mathcal{F}_d) = \mathcal{R}(n, 0, \mathcal{F}_g, \mathcal{F}_c, \mathcal{F}_d)$ to denote the corresponding uncontaminated rates derived by Uppal et al. [2019]. They used the wavelet thresholding estimator, proposed in Donoho et al. [1996], to derive a minimax convergence rate of the form

$$\mathcal{R}\left(n, \mathcal{F}_g, \mathcal{F}_d\right) = n^{-1/2} + n^{-\frac{\sigma_g + \sigma_d}{2\sigma_g + D}} + n^{-\frac{\sigma_g + \sigma_d - D/p_g + D/p_d'}{2\sigma_g + D(1 - 2/p_g)}}, \tag{8}$$

(omitting polylog factors in $n$). Extending a classical result of Donoho et al. [1996], they also showed that, if the estimator is restricted to be linear (in the sense of Def. 2), then the minimax rate slows to

$$\mathcal{R}_L\left(n, \mathcal{F}_g, \mathcal{F}_d\right) = n^{-1/2} + n^{-\frac{\sigma_g + \sigma_d}{2\sigma_g + D}} + n^{-\frac{\sigma_g + \sigma_d - D/p_g + D/p_d'}{2\sigma_g + D(1 - 2/p_g + 2/p_d')}}. \tag{9}$$

The first two terms in (8) & (9) are identical and the third term in (9) is slower. In particular, when $p_d' > p_g$ and $\sigma_d < D/2$, linear estimators are strictly sub-optimal, while the wavelet thresholding estimator converges at the optimal rate. The present paper extends this work in two directions.

First, we study how the minimax risk of estimating the data density $p$ changes when the observed data are contaminated by a proportion $\epsilon$ of outliers from a (potentially adversarially chosen) contamination distribution $g$. We show that, in most cases, wavelet thresholding estimators remain minimax optimal under both structured and unstructured contamination settings. Moreover, for $p_d' \le p_g$ linear wavelet estimators are minimax optimal under the structured contamination setting and the unstructured contamination setting if the IPM is generated by a smooth enough class of functions ($\sigma_d \ge D/p_d$).

Second, noting that the estimators of Uppal et al. [2019] rely on knowledge of the smoothness parameter $\sigma_g$ of the true density, we consider the more realistic case where $\sigma_g$ is unknown. We develop a fully data-dependent variant of the wavelet thresholding estimator from Uppal et al. [2019] that is minimax optimal for all $\sigma_g$ under structured contamination.

Finally, Uppal et al. [2019] also applied their results to bound the risk of a particular generative adversarial network (GAN) architecture. They show that the GAN is able to learn Besov densities at the minimax optimal rate. In this paper, we show that the same GAN architecture continues to be minimax optimal in the presence of outliers, and that, with minor modifications, it can do so without knowledge of the smoothness $\sigma_g$ of the true density.

### 3.2 Nonparametric Density Estimation with Huber Contamination

Chen et al. [2016] give a unified study of a large class of robust nonparametric estimation problems under the total variation loss. In the particular case of estimating a $\sigma_g$-Hölder continuous density,

their results imply a minimax convergence rate of $n^{-\frac{\sigma_g}{2\sigma_g+1}} + \epsilon$, matching our results (theorem 3) for total variation loss. The results of Chen et al. [2016] are quite specific to total variation loss, whereas, we provide results for a range of loss functions as well as densities of varying smoothness. Moreover, the estimator studied by Chen et al. [2016] is not computable in practice. It involves solving a testing problem between all pairs of points in a total variation cover of the hypothesis class in which the true density is assumed to lie. In contrast, our upper bounds rely on a simple thresholded wavelet series estimator, which can be computed in linear time (in the sample size $n$) with a fast wavelet transform.

Liu and Gao [2017] studied 1-dimensional density estimation at a point $x$ (i.e., estimating $p(x)$ instead of the entire density $p$) for Hölder smoothness densities under the Huber $\epsilon$-contamination model. In the case of unstructured contamination (arbitrary $G$), Liu and Gao [2017] derived a minimax rate of

$$n^{-\frac{\sigma_0}{2\sigma_0+1}} + \epsilon^{\frac{\sigma_0}{\sigma_0+1}} \tag{10}$$

in root-mean-squared error. With the caveats that we study estimation of the entire density $p$ rather than a single point $p(x)$ and assume that $G$ has a density $g$, this corresponds to our setting when $p_g = q_g = \infty$, and $D = 1$. Our results (equation 18) imply an upper bound on the rate of

$$n^{-\frac{\sigma_0}{2\sigma_0+1}} + \epsilon^{\frac{\sigma_0}{\sigma_0+(1-1/p)}} \tag{11}$$

under the $\mathcal{L}^p$ loss. Interestingly, this suggests that estimating a density at a point under RMSE is harder than estimating an entire density under $\mathcal{L}^2$ loss, and is, in fact, as hard as estimation under $\mathcal{L}^\infty$ (sup-norm) loss. While initially perhaps surprising, this makes sense if one thinks of rates under $\mathcal{L}^\infty$ loss as being the rate of estimating the density at the worst-case point over the sample space, which may be the point $x$ at which Liu and Gao [2017] estimate $p(x)$; under minimax analysis, these become similar.

We generalize these rates to (a) dimension $D > 1$, (b) densities $p$ lying in Besov spaces $B_{p_g,q_g}^{\sigma_g}$, and (c) a wide variety of losses parametrized by Besov IPMs ($B_{p_d,q_d}^{\sigma_d}$).

Liu and Gao [2017] also study the case of structured contamination, in which $g$ is assumed to be $\sigma_c$-Hölder continuous. Because they study estimation at a point, their results depend on an additional parameter, denoted $m$, which bounds the value of the contamination density $g$ at the target point (i.e., $g(x) \le m$). They derive a minimax rate of

$$n^{-\frac{\sigma_g}{2\sigma_g+1}} + \epsilon \min\{1, m\} + n^{-\frac{\sigma_c}{2\sigma_c+1}} \epsilon^{-\frac{\sigma_c}{2\sigma_c+1}}. \tag{12}$$

This rate contains a term depending only on $n$ that is identical to the minimax rate in the uncontaminated case, a term depending only on $\epsilon$, and a third "mixed" term. Notably, one can show that this mixed term $n^{-\frac{\sigma_c}{2\sigma_c+1}} \epsilon^{-\frac{\sigma_c}{2\sigma_c+1}}$ is always dominated by $n^{-\frac{\sigma_g}{2\sigma_g+D}} + \epsilon$, and so, unless $m \to 0$ as $n \to \infty$, the mixed term is negligible. In this paper, because we study estimation of the entire density $p$, the role of the parameter $m$ is played by $M := \|g\|_\infty$. Since $g$ is assumed to be a density with bounded support, we cannot have $M \to 0$; thus, in our results, the mixed term does not appear. Aside from this distinction, our results (Theorem 5) again generalize the results of Liu and Gao [2017] to higher dimensions, other Besov classes of densities, and new IPM losses.

Finally, we mention two early papers on robust nonparametric density estimation by Kim and Scott [2012] and Vandermeulen and Scott [2013]. These papers introduced variants of kernel density estimation based on $M$-estimation, for which they demonstrated robustness to arbitrary contamination using influence functions. These estimators are more complex than the scaled series estimates we consider, in that they non-uniformly weight the kernels centered at different sample points. While they also showed $\mathcal{L}^1$ consistency of these estimators, they did not provide rates of convergence, and so it is not clear when these estimators are minimax optimal.

## 4 Minimax Rates

Here we give our main minimax bounds. First, we state the estimators used for the upper bounds.

**Estimators:** To illustrate the upper bounds we consider two estimators that have been widely studied in the uncontaminated setting (see Donoho et al. [1996], Uppal et al. [2019]) namely the wavelet thresholding estimator and the linear wavelet estimator. All bounds provided here are tight up to polylog factors of $n$ and $1/\epsilon$.

For any $j_1 \geq j_0 \geq 0$ the wavelet thresholding estimator is defined as

$$\widehat{p}_n = \sum_{\phi \in \Phi} \widehat{\alpha}_\phi \phi + \sum_{j=0}^{j_0} \sum_{\psi \in \Psi_j} \widehat{\beta}_\psi \psi + \sum_{j=j_0}^{j_1} \sum_{\psi \in \Psi_j} \widetilde{\beta}_\psi \psi \tag{13}$$

where $\widehat{\alpha}_\phi = \frac{1}{n}\sum_{i=1}^n \phi(X_i)$ and $\widehat{\beta}_\psi = \frac{1}{n}\sum_{i=1}^n \psi(X_i)$ and coefficients of some of the wavelets with higher resolution (i.e., $j \in [j_0, j_1]$) are hard-thresholded: $\widetilde{\beta}_\psi = \widehat{\beta}_\psi 1_{\widehat{\beta}_\psi \geq t}$ for threshold $t = c\sqrt{j/n}$, where $c$ is a constant.

The linear wavelet estimator is simply $\widehat{p}_n$ with only linear terms (i.e., $j_0 = j_1$). Here $j_0, j_1$ correspond to smoothing parameters which we carefully choose to provide upper bounds on the risk. In the sequel, let $\mathcal{F}_g = B_{p_g, q_g}^{\sigma_g}(L_g)$ and $\mathcal{F}_d = B_{p_d, q_d}^{\sigma_d}(L_d)$ be Besov spaces.

## 4.1 Unstructured Contamination

In this section we consider the density estimation problem under Huber's $\epsilon$ contamination model; i.e. we have no structural assumptions on the contamination. Let $X_1, \ldots, X_n \stackrel{IID}{\sim} (1-\epsilon)p + \epsilon g$, where $p$ is the true density and $g$ is any compactly supported probability density. We provide bounds on the minimax risk of estimating the density $p$. We let

$$\mathcal{M}(\epsilon, \mathcal{F}_g) = \{(1-\epsilon)p + \epsilon g : p \in \mathcal{F}_g, g \text{ has compact support}\} \tag{14}$$

and bound the minimax risk

$$\mathcal{R}(n, \epsilon, \mathcal{F}_g, \mathcal{F}_d) = \inf_{\widehat{p}} \sup_{f \in \mathcal{M}(\epsilon, \mathcal{F}_g)} \mathbb{E}_f \, d_{\mathcal{F}_d}(\widehat{p}, p) \tag{15}$$

where the infimum is taken over all estimators $\widehat{p}_n$ constructed from the $n$ IID samples.

We first present our results for what Uppal et al. [2019] called the "Sparse" regime $p_d' \geq p_g$, in which the worst-case error is caused by large "spikes" in small regions of the sample space. Within this "Sparse" regime, we are able to derive minimax convergence rates for all Besov spaces $\mathcal{F}_g = B_{p_g, q_g}^{\sigma_g}(L_g)$ and $\mathcal{F}_d = B_{p_d, q_d}^{\sigma_d}(L_d)$. Surprisingly, we find that linear and nonlinear estimators have identical, rate-optimal dependence on the contamination proportion $\epsilon$ in this setting. Consequently, if $\epsilon$ is sufficiently large, then the difference in asymptotic rate between linear and nonlinear estimators vanishes. We first show the minimax rate in this setting that is achieved by a scaled version wavelet thresholding estimator i.e. $\frac{1}{(1-\epsilon)}\widehat{p}_n$. The proof is provided in section B.1 of the appendix.

**Theorem 3.** *(Minimax Rate, Sparse Case)* Let $r > \sigma_g > D/p_g$ and $p_d' \geq p_g$. Then,

$$\mathcal{R}\left(n, \epsilon, B_{p_g, q_g}^{\sigma_g}, B_{p_d, q_d}^{\sigma_d}\right) \sim \mathcal{R}\left(n, B_{p_g, q_g}^{\sigma_g}, B_{p_d, q_d}^{\sigma_d}\right) + \epsilon + \epsilon^{\frac{\sigma_g + \sigma_d + D/p_d' - D/p_g}{\sigma_g - D/p_g + D}} \tag{16}$$

On the other hand linear estimators are only able to achieve the following asymptotic rate. The proof is provided in section B.2 of the appendix.

**Theorem 4.** *(Linear Minimax Rate, Sparse Case)* Let $r > \sigma_g > D/p_g$ and $p_d' \geq p_g$. Then,

$$\mathcal{R}_L\left(n, \epsilon, B_{p_g, q_g}^{\sigma_g}, B_{p_d, q_d}^{\sigma_d}\right) \sim \mathcal{R}_L\left(n, B_{p_g, q_g}^{\sigma_g}, B_{p_d, q_d}^{\sigma_d}\right) + \epsilon + \epsilon^{\frac{\sigma_g + \sigma_d + D/p_d' - D/p_g}{\sigma_g - D/p_g + D}} \tag{17}$$

As is expected, the sub-optimality of linear estimators referred to in section 3, extends to the contaminated setting when contamination $\epsilon$ is small. However, if the contamination $\epsilon$ is large the distinction between linear and non-linear estimators disappears. More specifically, if $\epsilon$ is large enough then both estimators converge at the same rate of $\epsilon + \epsilon^{\frac{\sigma_g + \sigma_d + D/p_d' - D/p_g}{\sigma_g - D/p_g + D}}$.

**Bounds for the regime $p_d' \leq p_g$:** We note that the lower bounds that constitute the minimax rates above hold for all values of $p_g, p_d' \geq 1$. Furthermore, the linear wavelet estimator implies an upper bound (shown in section B.3 of the appendix) on the risk in the dense regime. Together, this gives, for all $r > \sigma_g > D/p_g$ and $p_d' \leq p_g$,

$$\Delta(n) + \epsilon + \epsilon^{\frac{\sigma_g + \sigma_d + D/p_d' - D/p_g}{\sigma_g - D/p_g + D}} \leq \mathcal{R}\left(n, \epsilon, B_{p_g, q_g}^{\sigma_g}, B_{p_d, q_d}^{\sigma_d}\right) \leq \Delta(n) + \epsilon + \epsilon^{\frac{\sigma_g + \sigma_d}{\sigma_g + D/p_d}} \tag{18}$$

where $\Delta(n) = \mathcal{R}\left(n, B_{p_g, q_g}^{\sigma_g}, B_{p_d, q_d}^{\sigma_d}\right)$.

One can check that, when the discriminator is sufficiently smooth (specifically, $\sigma_d \geq D/p_d$), the term $\epsilon^{\frac{\sigma_g + \sigma_d}{\sigma_g + D/p_d}}$ on the right-hand side of Eq. (18) is dominated by $\epsilon$; hence, the lower and upper bounds in Eq. (18) match and the thresholding wavelet estimator is minimax rate-optimal. When $\sigma_d < D/p_d$, a gap remains between our lower and upper bounds, and we do not know whether the thresholding wavelet estimator is optimal. The sample mean is generally well-known to be sensitive to outliers in the data, and a large amount of recent work [Lugosi and Mendelson, 2016, Lerasle et al., 2018, Minsker et al., 2019, Diakonikolas et al., 2019] has proposed estimators that might be better predictors of the mean in the case of contamination by outliers. Since the linear and thresholding wavelet estimators are both functions of the empirical means $\widehat{\beta}_\psi$ of the wavelet basis functions, we conjecture that a density estimator based on a better estimate of the wavelet mean $\beta_\psi^p$ might be able to converge at a faster rate as $\epsilon \to 0$. We leave this investigation for future work.

## 4.2 Structured Contamination

In the previous section, we analyzed minimax rates without any assumptions on the outlier distribution. In certain settings, this may be an overly pessimistic contamination model, and the outlier distribution may in fact be somewhat well-behaved. In this section, we study the effects of assuming the contamination distribution $G$ has a density $g$ that is either bounded or smooth. Our results show that assuming boundedness of $g$ improves the dependence of the minimax rate on $\epsilon$ to order $\asymp \epsilon$, but assuming additional smoothness of $g$ does not further improve rates.

As described in Section 2, in this setting we consider a more general form of the minimax risk:

$$\mathcal{R}(n, \epsilon, \mathcal{F}_g, \mathcal{F}_c, \mathcal{F}_d) = \inf_{\widehat{p}} \sup_{f \in \mathcal{M}(\epsilon, \mathcal{F}_g, \mathcal{F}_c)} \mathbb{E}_f[d_{\mathcal{F}_d}(\widehat{p}, p)] \tag{19}$$

The additional parameter $\mathcal{F}_c$ denotes the class of allowed contamination distributions.

We provide the following asymptotic rate for the above minimax risk that is achieved by an adaptive wavelet thresholding estimator with $2^{j_0} = n^{\frac{1}{2r+D}}$ and $2^{j_1} = (n/\log n)^{1/D}$. Recall here that $r$ is the regularity of the wavelets. Thus, for any $\sigma_g < r$, this estimator does not require the knowledge of $\sigma_g$.

**Theorem 5** (Minimax Rate under Structured Contamination). *Let $\sigma_g \geq D/p_g$, $\sigma_c > D/p_c$ and $\epsilon \leq 1/2$. Then, up to poly logarithmic factors of $n$,*

$$\mathcal{R}\left(n, \epsilon, B_{p_g, q_g}^{\sigma_g}, B_{p_c, q_c}^{\sigma_c}, B_{p_d, q_d}^{\sigma_d}\right) \asymp \mathcal{R}\left(n, \epsilon, B_{p_g, q_g}^{\sigma_g}, \mathcal{L}^\infty, B_{p_d, q_d}^{\sigma_d}\right) \asymp \mathcal{R}\left(n, B_{p_g, q_g}^{\sigma_g}, B_{p_d, q_d}^{\sigma_d}\right) + \epsilon \tag{20}$$

The right-most term is simply $\epsilon$ plus the rate in the absence of contamination. The left two terms are the rates when the contamination density lies, respectively, in the Besov space $B_{p_c, q_c}^{\sigma_c}$ and the space $\mathcal{L}^\infty$ of essentially bounded densities. In particular, these rates are identical when $\sigma_c > D/p_c$. One can check (see Lemma 10 in the Appendix) that, if $\sigma_c > D/p_c$, then $B_{p_c, q_c}^{\sigma_c} \subseteq \mathcal{L}^\infty$. Hence, Theorem 5 shows that assuming boundedness of the contamination density improves the dependence on $\epsilon$ (compared to unstructured rates from the previous section), but that additional smoothness assumptions do not help.

In section B.1 of the appendix we first provided a proof of the upper bound using the classical wavelet thresholding estimator and then show the optimality of the adaptive version in section B.4.

## 4.3 Examples

Here, we summarize the implications of our main results for robust density estimation in a few specific examples, allowing us to directly compare with previous results.

The case $p_d = q_d = \infty$, includes, as examples, the total variation loss $d_{B_{p_d, q_d}^0}$ ($\sigma_d = 0$, Rudin [2006]) and the Wasserstein (a.k.a., Kantorovich-Rubinstein or earthmover) loss $d_{B_{p_d, q_d}^1}$ ($\sigma_d = 1$ [Villani, 2008]). Under these losses, the wavelet thresholding estimator is robustly minimax optimal, in both the arbitrary and structured contamination settings (note that here $\sigma_d \geq D/p_d = 0$). In particular, in the case of unstructured contamination, this generalizes the results of Chen et al. [2016] for total variation loss to a range of other losses and smoothness assumptions on $p$.

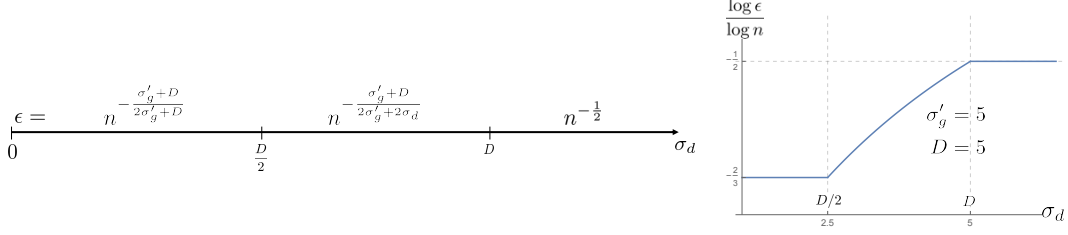

Figure 2: Asymptotic breakdown point as a function of $\sigma_d$, in the case $p_d = 1$; this includes as special cases the $\mathcal{L}^\infty$ and Kolmogorov-Smirnov losses.

Analogously, in the case $p_g = q_g = \infty$, the data distribution is itself $\sigma_g$-Hölder continuous, since the Besov space $B_{p_g,q_g}^{\sigma_g} = \mathcal{C}^{\sigma_g}$ is equivalent to the space of $\sigma_g$-Hölder continuous functions. In this setting, the linear wavelet estimator is robustly minimax optimal under any Besov IPM loss when contamination is structured, or under sufficiently smooth Besov IPM losses (with $\sigma_d \geq D/p_d$) when the contamination is unstructured.

One can also use our results to calculate the sensitivity of a given estimator to the proportion $\epsilon$ of outlier samples. In the terminology of robust statistics, this is quantified by the "asymptotic breakdown point" (i.e., the maximum proportion $\epsilon$ of outlier samples such that the estimator can still converge at the uncontaminated optimal rate). Figure 2 illustrates the asymptotic breakdown point, in the case $p_d = 1$, as a function of the discriminator smoothness $\sigma_d$. For sufficiently smooth losses (large $\sigma_d$, the estimator can tolerate a large number ($O(\sqrt{n})$) of arbitrary outliers before performance begins to degrade, whereas, for stronger losses (smaller $\sigma_d$), the estimator becomes more sensitive to outliers.

## 5   Robustness of Generative Adversarial Networks

Singh et al. [2018] showed (in their Theorem 9) that the problems of generating novel samples from a training density (also called "implicit generative modeling" [Mohamed and Lakshminarayanan, 2016]) and of estimating the training density are equivalent in terms of statistical minimax rates. Based on this result, and an oracle inequality of Liang [2018], several recent works [Liu et al., 2017, Liang, 2018, Singh et al., 2018, Uppal et al., 2019] have studied a statistical formulation of GANs as a distribution estimate based on empirical risk minimization (ERM) under an IPM loss. This formulation is as follows. Given a GAN with a discriminator neural network $N_d$ encoding functions in $\mathcal{F}$ and a generator neural network $N_g$ encoding distributions in $\mathcal{P}$, the GAN generator can be viewed as the distribution $\widehat{P}$ satisfying:

$$\widehat{P} = \inf_{P \in \mathcal{P}} d_{\mathcal{F}}(P, \widetilde{P}_n) \tag{21}$$

While $\widetilde{P}_n$ can be taken to be the empirical distribution $\frac{1}{n} \sum_{i=1}^{n} \delta_{X_i}$, these theoretical works have shown that convergence rates can be improved by applying regularization (e.g., in the form of smoothing the empirical distribution), consistent with the "instance noise trick" [Sønderby et al., 2016], a technique that is popular in practical GAN training and is mathematically equivalent to kernel smoothing. Here, we extend these results to the contamination setting and show that the wavelet thresholding estimator can be used to construct a GAN estimate that is robustly minimax optimal.

Suzuki [2018] (in section 3) showed that there is a fully connected ReLU network with depth at most logarithmic in $1/\delta$ and other size parameters at most polynomial in $1/\delta$ that can $\delta$-approximate any sufficiently smooth Besov function class (e.g. $B_{p_g,q_g}^{\sigma_g}$ with $\sigma_g \geq D/p_g$). This was used in Uppal et al. [2019] to show that, for large enough network sizes, the perfectly optimized GAN estimate (of the form of Eq. (21)) converges at the same rate as the estimator $\widehat{p}$ used to generate it. So if we let the approximation error of the generator and discriminator network be at most the convergence rate (from Theorem 3 or Theorem 5) of the wavelet thresholding estimator then there is a GAN estimate $\widehat{P}$ that converges at the same rate and is therefore robustly minimax optimal. In particular, we have the following corollary:

**Corollary 6.** *Given a Besov density class $B_{p_g,q_g}^{\sigma_g}$ with $\sigma_g > D/p_g$ and discriminator class $B_{p_d,q_d}^{\sigma_d}$ with $\sigma_d > D/p_d$, there is a GAN estimate $\widehat{P}$ with discriminator and generator networks of depth at most logarithmic in $n$, or $1/\delta$ and other size parameters at most polynomial in $n$ or $1/\delta$ such that*

$$\sup_{p \in \mathcal{M}\left(\epsilon, B_{p_g,q_g}^{\sigma_g}\right)} \mathbb{E}\left[d_{B_{p_d,q_d}^{\sigma_d}}(\widehat{p}, p)\right] \leq \delta + \mathcal{R}\left(n, B_{p_g,q_g}^{\sigma_g}, B_{p_d,q_d}^{\sigma_d}\right) + \epsilon \tag{22}$$

Since Besov spaces of compactly supported densities are nested ($B_{p,q}^{\sigma} \subseteq B_{p,q}^{\sigma'}$ for all $\sigma' \leq \sigma$), to approximate $B_{p,q}^{\sigma}$ for any $\sigma \geq r_0$ it is sufficient to approximate $B_{p,q}^{r_0}$. We can use this approximation network along with the adaptive wavelet thresholding estimator to construct a GAN estimate of the form of Eq. (21). Then under structured contamination this GAN estimate is minimax optimal for any density of smoothness $\sigma_g \in [r_0, r]$ and does not require explicit knowledge of $\sigma_g$. Thus, it is adaptive.

# 6 Conclusion

In this paper, we studied a variant of nonparametric density estimation in which a proportion of the data are contaminated by random outliers. For this problem, we provided bounds on the risks of both linear and nonlinear wavelet estimators, as well as general minimax rates. The main conclusions of our study are as follows:

1. The classical wavelet thresholding estimator originally proposed by Donoho et al. [1996], which is widely known to be optimal for uncontaminated nonparametric density estimation, continues to be, in many settings, minimax optimal in the presence of contamination.

2. Imposing a simple structural assumption, such as bounded contamination, can significantly alter how contamination affects estimation risk. At the same time, additional smoothness assumptions have no effect. This contrasts from the case of estimating a density at a point, as studied by Liu and Gao [2017] where the minimax rates get better with smoothness of the contamination density.

3. Linear estimators, exhibit optimal dependence on the contamination proportion, despite having sub-optimal risk with respect to the sample size. Hence, the difference between linear and nonlinear models diminishes in the presence of significant contamination.

4. For sufficiently smooth density and discriminator class, a fully-connected GAN architecture with ReLU activations can learn the distribution of the training data at the optimal rate, both (a) in the presence of contamination and (b) when the true smoothness of the density is not known.

Our results both extend recent results on nonparametric density estimation under IPM losses [Liang, 2018, Singh et al., 2018, Uppal et al., 2019] to the contaminated and adaptive settings and expand the study of nonparametric density estimation under contamination [Chen et al., 2016, Liu and Gao, 2017] to Besov densities and IPM losses.

## Broader Impact

Since this work is of a theoretical nature, it is unlikely to disadvantage anyone or otherwise have significant negative consequences. One of the main contributions of this paper is to quantify the potential effects of misspecification biases on density estimation. Hence, the results in this paper may help researchers understand the potential effects of misspecification biases that can arise when invalid assumptions are made about the nature of the data generating process.

## Acknowledgments and Disclosure of Funding

The authors thank anonymous reviewers for the feedback on improving this paper. The authors declare no competing interests. This work was supported by National Science Foundation award number DGE1745016, a grant from JPMorgan Chase Bank, and a grant from the Lockheed Martin Corporation.

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
