[Supplementary Material]

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

# A Set up

Besov spaces rely on the notion of an $r$-regular multi-resolution approximation (MRA) of $\mathcal{L}_2(\mathbb{R}^D)$. In particular, the father wavelet of the wavelet basis used to define Besov spaces generates an MRA of $\mathcal{L}_2(\mathbb{R}^D)$.

The goal of a MRA is to efficiently approximate spatially varying smoothness. Härdle et al. [2012] explains it as a formulation that makes mathematically precise the intuitive idea of partitioning the domain and applying Fourier analysis to each piece.

Here we formally define an $r$-regular multi-resolution approximation.

**Definition 7.** *A* multiresolution approximation (MRA) *of $\mathcal{L}^2(\mathbb{R}^D)$ is a nested sequence $\{V_j\}_{j \in \mathbb{Z}}$ of closed linear subspaces of $L^2(\mathbb{R}^D)$ such that:*

1. *$\bigcap_{j=-\infty}^{\infty} V_j = \{0\}$, and $\bigcup_{j=-\infty}^{\infty} V_j$ is dense in $\mathcal{L}^2(\mathbb{R}^D)$.*
2. *For every $f \in \mathcal{L}^2(\mathbb{R}^D)$ and $k \in \mathbb{Z}^D$, $f(x) \in V_0$ if and only if $f(x-k) \in V_0$.*
3. *For every $f \in \mathcal{L}^2(\mathbb{R}^D)$ and $j \in \mathbb{Z}$, $f(x) \in V_j$ if and only if $f(2x) \in V_{j+1}$*
4. *There is a "father wavelet" such that $\phi \in V_0$, $\{\phi(x-k) : k \in \mathbb{Z}^D\}$ is an orthonormal basis of $V_0 \subset \mathcal{L}^2(\mathbb{R}^D)$.*

Given a father wavelet that generates a multi-resolution approximation, there exist "mother" wavelets with the following properties.

**Lemma 8** ([Meyer, 1992], Section 3.9). *Let $\{V_j\}_{j \in \mathbb{Z}}$ be an MRA of $\mathcal{L}^2(\mathbb{R}^D)$ with father wavelet $\phi$. , and let $W_j$ be the orthogonal complement of $V_j$ in $V_{j+1}$. Then, for $E = \{0,1\}^D \setminus (0, \ldots, 0)$, there exist "mother wavelets" $\{\psi_\epsilon\}_{\epsilon \in E}$ such that*

1. *$\psi_\epsilon$ is rapidly decreasing for every multi-index $\alpha$ with $|\alpha| \leq r$ and every $\epsilon \in E$.*
2. *The set $\{\psi(x-k)\}_{\epsilon \in E, k \in \mathbb{Z}}$ is an orthonormal basis of $W_j$.*
3. *For all $\alpha$ with $|\alpha| \leq r$ and $\epsilon \in E$, $\int x^\alpha \psi_\epsilon(x) dx = 0$.*

*Moreover, $\{2^{Dj/2}\psi_\epsilon(2^j x - k) : \epsilon \in E, k \in \mathbb{Z}^D\} \cup \{2^{Dj/2}\phi(2^j x - k) : k \in \mathbb{Z}^D\}$ is an orthonormal basis of $V_j \subseteq \mathcal{L}^2(\mathbb{R}^D)$.*

The $r$-regularity of the mother wavelet as described part 3 of the above lemma determines the $r$-regularity of the wavelet basis.

# B Upper Bounds

## B.1 Non-Linear Rate

In this section we provide proofs of the upper bounds stated above under both structured and unstructured contamination i.e. theorems 3 and 5. We will use a scaled version of the wavelet thresholding estimator to demonstrate these results. The proofs follow along the same lines as those of the uncontaminated version except the usual bias-variance trade-off now has an additional term; the misspecification error.

In particular, the bound on the bias remains unchanged. Moreover, we show that for resolutions small enough the variance can be bounded by the same term as before. This is straightforward for the variance of the linear terms but somewhat involved for that of the non-linear terms. So, we derive the bound for the non-linear terms at the very end.

There is a qualitative difference between the misspecification error under the structured and unstructured settings. When the contamination density is bounded, the misspecification error is simple bounded by the contamination proportion $\epsilon$; in the unstructured setting, this error depends on the number of terms considered in the estimator.

We now provide the formal proof. Let

$$\mathcal{P} = \{p : p \geq 0, \|p\|_{\mathcal{L}^1} = 1, \text{supp}(p) \subseteq [-T, T]\}$$

denote the set of densities that are supported on the interval $[-T, T]$. We have assumed that our discriminator and generator classes are, respectively,

$$\mathcal{F}_d = \{f : \|f\|_{p_d, q_d}^{\sigma_d} \le L_d\}$$
$$\text{and} \quad \mathcal{F}_g = \{p : \|p\|_{p_g, q_g}^{\sigma_g} \le L_g\} \cap \mathcal{P}.$$

Let for any density function $p$

$$\alpha_\phi^p = \underset{X \sim p}{\mathbb{E}}[\phi(X)]$$
$$\text{and} \quad \beta_\psi^p = \underset{X \sim p}{\mathbb{E}}[\psi(X)].$$

Since $p \in \mathcal{F}_g$, we have that

$$p = \sum_{\phi \in \Phi} \alpha_\phi^p \phi + \sum_{j \ge 0} \sum_{\psi \in \Psi_j} \beta_\psi^p \psi,$$

where the convergence is in the $L_p$ norm.

For the unstructured setting we merely assume that the contamination density is compactly supported on $[-T, T]$. Under the structured contamination setting, we additionally assume that the contamination density $g$ is essentially bounded i.e. $\mathcal{F}_c = \mathcal{L}^\infty(L_c)$ (where $L_c$ is a uniform bound on the $\mathcal{L}_\infty$ norm of any $g \in \mathcal{F}_c$).

We first show that it is enough to consider the "sparse" case (so called by Donoho et al. [1996]) characterized by $p_d' \ge p_g$ by the following lemma.

**Lemma 9.** *For $p_d' \le p_g$ and compactly supported densities $p, q \in \mathcal{L}_{p_g} \subseteq \mathcal{L}_{p_d'}$ we have that,*

$$d_{\mathcal{B}_{p_d, q_d}^{\sigma_d}}(p, q) \le d_{\mathcal{B}_{p_g', q_d}^{\sigma_d}}(p, q).$$

*Proof.* Suppose $p, q$ are compactly supported on $[-T, T]$ then it is enough to show that

$$\mathcal{B}_{p_d, q_d}^{\sigma_d}(T) \subseteq \mathcal{B}_{p_g', q_d}^{\sigma_d}(T).$$

Using the fact that $p_d \ge p_g'$, this is clear since,

$$2^{j(\sigma_d + D/2 - D/p_g')} \|\beta_j\|_{p_g'} \le 2^{j(\sigma_d + D/2 - D/p_d)} \|\beta_j\|_{p_d}$$

by using the simple fact that for a $2^{Dj}$-dimensional vector $x$, $\|x\|_{p_g'} \le 2^{Dj(1/p_g' - 1/p_d)} \|x\|_{p_d}$. $\quad\square$

Let $\widehat{p}_n$ be the wavelet thresholding estimator of $p$ introduced by Donoho et al. [1996];

$$\widehat{p}_n = \sum_{\phi \in \Phi} \widehat{\alpha}_\phi \phi + \sum_{j=0}^{j_0} \sum_{\psi \in \Psi_j} \widehat{\beta}_\psi \psi + \sum_{j=j_0}^{j_1} \sum_{\psi \in \Psi_j} \widetilde{\beta}_\psi \psi$$

where we threshold the higher resolution terms i.e.

$$\alpha_\phi^p = \underset{X \sim p}{\mathbb{E}}[\phi(X)] \qquad \widehat{\alpha}_\phi = \frac{1}{n} \sum_{i=1}^n \phi(X_i)$$
$$\beta_\psi^p = \underset{X \sim p}{\mathbb{E}}[\psi(X)] \qquad \widehat{\beta}_\psi = \frac{1}{n} \sum_{i=1}^n \psi(X_i)$$
$$\widetilde{\beta}_\psi = \widehat{\beta}_\psi \mathbf{1}_{\{\widehat{\beta}_\psi > t\}}$$

with threshold $t = K\sqrt{j/n}$, where $K$ is a constant to be specified later, and

$$2^{j_0} = \sqrt{n}^{\frac{1}{\sigma_g + D/2}}$$
$$2^{j_1} = \sqrt{n}^{\frac{1}{\sigma_g + D/2 - D/p_g}} \wedge \epsilon^{-\frac{1}{\sigma_g + D - D/p_g}}$$

We will use a scaled version of this estimator i.e. $\frac{1}{1-\epsilon} \widehat{p}_n$.

We decompose the risk of the above estimator as follows. At each resolution $\widehat{\alpha}_\phi$ or $\widehat{\beta}_\psi$ is an unbiased estimate of the co-efficient of the contaminated density $(1-\epsilon)p + \epsilon g$. So, by the triangle inequality, we can decompose the error as

$$\mathbb{E}\, d_\mathcal{F}\left(\frac{\widehat{p}_n}{1-\epsilon}, p\right)$$

$$\leq \frac{1}{1-\epsilon}\, \mathbb{E}\, d_\mathcal{F}\left(\sum_{\phi\in\Phi}\widehat{\alpha}_\phi\phi, \sum_{\phi\in\Phi}(\alpha_\phi^p + \epsilon\alpha_\phi^g)\phi\right) \tag{23}$$

$$+ \frac{1}{1-\epsilon}\, d_\mathcal{F}\left(\sum_{j=0}^{j_0}\sum_{\psi\in\Psi_j}\widehat{\beta}_\psi\psi, \sum_{j=0}^{j_0}\sum_{\psi\in\Psi_j}(\beta_\psi^p + \epsilon\beta_\psi^g)\psi\right) \tag{24}$$

$$+ \frac{1}{1-\epsilon}\, d_\mathcal{F}\left(\sum_{j=j_0}^{j_1}\sum_{\psi\in\Psi_j}\widetilde{\beta}_\psi\psi, \sum_{j=j_0}^{j_1}\sum_{\psi\in\Psi_j}(\beta_\psi^p + \epsilon\beta_\psi^g)\psi\right) \tag{25}$$

$$+ \frac{1}{1-\epsilon}\, d_\mathcal{F}\left(\sum_{\phi\in\Phi}\alpha_\phi^p\phi + \sum_{j=0}^{j_1}\sum_{\psi\in\Psi_j}\beta_\psi^p\psi, p\right) \tag{26}$$

$$+ \epsilon d_\mathcal{F}\left(\sum_{\phi\in\Phi}\alpha_\phi^g\phi + \sum_{j=0}^{j_1}\sum_{\psi\in\Psi_j}\beta_\psi^g\psi, 0\right) \tag{27}$$

where the first two terms constitute the error of the linear terms, the third term is the error of the non-linear terms, the fourth term is the bias and the last term is the misspecification error, respectively.

We will use the following upper bounds on the bias and variance of a linear wavelet estimator (when $j_0 = j_1$ above) from Appendix C of Uppal et al. [2019].

First we see that under Besov IPMs, if the moments of the wavelet co-efficients of the density don't grow too fast with the resolution then the variance of the linear wavelet estimator can be conveniently bounded.

**Lemma 10. (Variance)** *Let* $X_1, \ldots, X_n \sim p$ *where* $p$ *is compactly supported and* $\mathcal{F}_d = B_{p_d, q_d}^{\sigma_d}$. *If* $\mathbb{E}_p |\psi(X)|^{p'_d} \leq c_{p'_d} 2^{Dj(p'_d/2-1)}$ *for all* $\psi \in \Psi_j$, *then the variance of a linear wavelet estimator* $\widehat{p}_n$ *with* $j_0$ *terms i.e.*

$$\widehat{p}_n = \sum_{\phi\in\Phi}\widehat{\alpha}_\phi\phi + \sum_{j=0}^{j_0}\sum_{\psi\in\Psi_j}\widehat{\beta}_\psi\psi$$

*is bounded by*

$$d_{\mathcal{F}_d}(\widehat{p}_n, \mathbb{E}[\widehat{p}_n]) \leq c\left(\frac{1}{\sqrt{n}} + \frac{2^{j_0(D/2-\sigma_d)}}{\sqrt{n}}\right)$$

*where* $c = c_{p'_d}\left(\mathbb{E}_p |\psi(X)|^2\right)^{1/2}$ *is a constant.*

Note here that we do not need the density to lie in a Besov space but to simply have the given bound on the moments of its wavelet coefficients. However, for a bound on the bias provided below we need the full power of the Besov space.

**Lemma 11. (Bias)** *Let* $X_1, \ldots, X_n \sim p$ *where* $p \in B_{p_g, q_g}^{\sigma_g}$ *is compactly supported and* $\sigma_g \geq D/p_g$, $\mathcal{F}_d = B_{p_d, q_d}^{\sigma_d}$. *Then the bias of a linear wavelet estimator* $\widehat{p}$ *with* $j_0$ *terms is bounded by*

$$d_{\mathcal{F}_d}(p, \mathbb{E}_p[\widehat{p}_n]) \leq c 2^{-j_0(\sigma_d + \sigma_g - (D/p_g - D/p'_d)_+)}$$

*where* $c = L_d L_g$ *is a constant.*

We will also need the following bound on a density living in a Besov space.

**Lemma 12. (Upper Bound on Smooth Besov Spaces)** *Let* $f \in B_{p_g, q_g}^{\sigma_g}$ *where* $\sigma_g > D/p_g$ *then*

$$\|f\|_\infty \leq 4A\, \|\psi\|_\infty\, L_g(1 - 2^{(\sigma_g - D/p_g)q'_g})^{-1/q'_g}$$

This lemma implies that sufficiently smooth Besov spaces $B_{p_g,q_q}^{\sigma_g}$ are uniformly bounded.

We are now ready to provide upper bounds on the risk in both the structured and unstructured setting whenever $p_d' \geq p_g$.

Under both the structured and unstructured contamination setting we can immediately bound the bias term using lemma 11 (since this is just the bias of a linear wavelet estimator with $j_1$ terms) by

$$2^{-j_1(\sigma_g + \sigma_d + D/p_d' - D/p_g)}$$

Since, $\sigma_g > D/p_g$ we know that by lemma 12, $\|p\|_\infty < \infty$. Therefore, for any $\psi \in \Psi_j$,

$$\underset{(1-\epsilon)p+\epsilon g}{\mathbb{E}} \left[ |\psi(X)|^{p_d'} \right] \leq (1-\epsilon) \|p\|_\infty 2^{Dj(p_d'/2-1)} + \epsilon \underset{g}{\mathbb{E}} \left[ |\psi(X)|^{p_d'} \right]$$

When contamination is structured i.e. $\|g\|_\infty < \infty$ we have

$$\epsilon \underset{g}{\mathbb{E}} \left[ |\psi(X)|^{p_d'} \right] \leq \epsilon \|g\|_\infty 2^{Dj(p_d'/2-1)}$$

and when the contamination density is not bounded above we have,

$$\epsilon \underset{g}{\mathbb{E}} |\psi(X)|^{p_d'} \leq \epsilon 2^{Djp_d'/2}$$

Since in this case, $2^{Dj_1} \leq \epsilon^{-\frac{D}{\sigma_g+D-D/p_g}} \leq 1/\epsilon$, $\epsilon \leq 2^{-Dj}$ for all $j \leq j_1$ the term above is always smaller than $2^{Dj(p_d'/2-1)}$.

So, under both cases, we have,

$$\underset{(1-\epsilon)p+\epsilon g}{\mathbb{E}} |\psi(X)|^{p_d'} \leq c 2^{Dj(p_d'/2-1)} \tag{28}$$

We can now use lemma 10 to bound the variance for both cases as

$$\mathbb{E}\, d_{\mathcal{F}}(\widehat{p}_n, \underset{(1-\epsilon)p+\epsilon g}{\mathbb{E}}[\widehat{p}_n]) \leq c \left( \frac{1}{\sqrt{n}} + \frac{2^{j_0(D/2-\sigma_d)}}{\sqrt{n}} \right)$$

We can also similarly bound the mis-specification error as this is simply the misspecification error of a linear wavelet estimator with $j_1$ terms. We then have an upper bound of

$$\epsilon 2^{j_1(D/p_d-\sigma_d)}.$$

We now bound the misspecification error i.e.

$$\frac{\epsilon}{1-\epsilon} d_{\mathcal{F}} \left( \underset{g}{\mathbb{E}}[\widehat{p}_n], 0 \right)$$

We will use the following lemmas proven by Uppal et al. [2019] to first reduce the expression of the above distance to one in terms of wavelet coefficients (of $g$) only.

**Lemma 13.** *Let $p$, $q$ be compactly supported probability densities and $\mathcal{F}_d = B_{p_d,q_d}^{\sigma_d}$, s.t. either $p, q \in L_{p_d'}$ or $\sigma_d > D/p_d$, then $d_{\mathcal{F}_d}(p, q) =$*

$$\underset{f \in \mathcal{F}_d}{\sup} \left| \sum_{\phi \in \Phi} \alpha_\phi^f \left( \alpha_\phi^p - \alpha_\phi^q \right) + \sum_{j \geq 0} \sum_{\psi \in \Psi_j} \beta_\psi^f \left( \beta_\psi^p - \beta_\psi^q \right) \right|$$

*where for $f \in \mathcal{F}_d$*

$$f = \sum_{\phi \in \Phi} \alpha_\phi^f \phi + \sum_{j \geq 0} \sum_{\psi \in \Psi_j} \beta_\psi^f \psi$$

**Lemma 14.** *Let $n_1, n_2 \in \mathbb{N} \cup \{\infty\}$ and $\eta$ be any sequence of numbers. Then*

$$\underset{X_1,\ldots,X_n}{\mathbb{E}} \underset{f \in \mathcal{F}_d}{\sup} \sum_{j=n_1}^{n_2} \sum_{\psi \in \Psi_j} \gamma_\psi^f \eta_\psi \leq L_d \sum_{j=n_1}^{n_2} 2^{-j\sigma_d'} \left( \underset{X_1,\ldots,X_n}{\mathbb{E}} \sum_{\psi \in \Psi_j} |\eta_\psi|^{p_d'} \right)^{1/p_d'}$$

*where $\sigma_d' = \sigma_d + D/2 - D/p_d$. Note that the above is true also if $\gamma = \alpha^f$ and $n_1 = n_2 = 0$.*

Applying the lemmas above we have for any contamination density $g$,

$$\epsilon d_{\mathcal{F}}\left(\underset{g}{\mathbb{E}}[\widehat{p}_n], 0\right) \leq c\epsilon \left(\|\alpha^g\|_{p'_d} + \sum_{j=0}^{j_1} 2^{-j(\sigma_d+D/2-D/p_d)} \|\beta_j^g\|_{p'_d}\right) \tag{29}$$

where for all $\phi \in \Phi$ and $\psi \in \Psi_j$, $\alpha_\phi^g = \int \phi(x)g(x)$ and $\beta_\psi^g = \int \psi(x)g(x)$.

When the contamination is structured we have the following upper bound on the wavelet coefficients

$$|\beta_\psi^g| = \left|\int \psi(x)g(x)d(x)\right| \leq \|\psi_\epsilon\|_\infty \|g\|_\infty 2^{-Dj/2} \implies \|\beta_\psi^g\|_{p'_d} \leq c2^{Dj(1/p'_d-1/2)} \tag{30}$$

where $\psi \in \Psi_j$ and $|\alpha_\phi^g| \leq \|\phi\|_\infty \|g\|_\infty$. Thus implying the following bound on 29

$$c\epsilon\left(1 + \sum_{j=0}^{j_0} 2^{-j(\sigma_d+D/2-D/p_d)} 2^{Dj/p'_d} 2^{-Dj/2}\right) \leq c\epsilon\left(1 + \sum_{j=0}^{j_0} 2^{-j\sigma_d}\right) \leq c\epsilon.$$

When the contamination is unstructured, by convexity we have,

$$\|\beta_j^g\|_{p'_d} = \left(\sum_{\psi \in \Psi_j} |\underset{g}{\mathbb{E}}\,\psi(X)|^{p'_d}\right)^{1/p'_d} \leq \left(\sum_{\psi \in \Psi_j} \underset{g}{\mathbb{E}}\,|\psi(X)|^{p'_d}\right)^{1/p'_d} \leq \left(\underset{g}{\mathbb{E}}\sum_{\psi \in \Psi_j} |\psi(X)|^{p'_d}\right)^{1/p'_d}$$

and we can interchange the expectation and sum in the last step because $g$ is compactly supported which implies there are only finitely many non-zero terms to sum. The compactness of the wavelets implies only finitely many wavelets overlap at a point. So we have,

$$\int \left(\sum_{\psi \in \Psi_j} |\psi(x)|^{p'_d}\right) g(x)dx \leq c2^{Djp'_d/2} \implies \|\beta_j^g\|_{p'_d} \leq c2^{Dj/2} \tag{31}$$

where $c$ might depend on the dimension. So we obtain the bound, (where the $\alpha$ term is bounded in the same way by a constant)

$$\epsilon d_{\mathcal{F}}\left(\underset{G}{\mathbb{E}}[\widehat{p}_n], 0\right) \leq c\epsilon\left(1 + \sum_{j=0}^{j_0} 2^{-j(\sigma_d+D/2-D/p_d)} 2^{Dj/2}\right) = c\epsilon\left(1 + \sum_{j=0}^{j_0} 2^{j(D/p_d-\sigma_d)}\right)$$

$$\leq c\epsilon\left(1 + 2^{j_0(D/p_d-\sigma_d)}\right)$$

So, it only remains to bound the risk of the non-linear terms i.e.

$$d_{\mathcal{F}}\left(\sum_{j=j_0}^{j_1} \sum_{\psi \in \Psi_j} \widetilde{\beta}_\psi \psi, \sum_{j=j_0}^{j_1} \sum_{\psi \in \Psi_j} (\beta_\psi^p + \epsilon\beta_\psi^g)\psi\right)$$

From lemmas 13 and 14 we will upper bound the following:

$$\sum_{j=j_0}^{j_1} 2^{-j(\sigma_d+D/2-D/p_d)} \left(\sum_{\psi \in \Psi_j} \mathbb{E}\,|(1-\epsilon)\beta_\psi^p + \epsilon\beta_\psi^g - \widetilde{\beta}_\psi|^{p'_d}\right)^{1/p'_d}$$

We will need the following moment and large deviation bounds from Uppal et al. [2019]:

**Lemma 15.** *(Moment Bounds) Let $X_1, \ldots, X_n \sim p$, $m \geq 1$ s.t. there is a constant $c$ with $\mathbb{E}_p\,|\psi(X)|^m \leq c2^{Dj(m/2-1)}$ for all $\psi \in \Psi_j$. Let*

$$\gamma_\psi^p = \mathbb{E}[\psi(X)],$$

$$\widehat{\gamma}_\psi = \frac{1}{n}\sum_{i=1}^n \psi(X_i),$$

*Then for all $j$ s.t. $2^{Dj} \in \mathcal{O}(n)$,*

$$\mathbb{E}[|\widehat{\gamma}_\psi - \gamma_\psi|^m] \leq cn^{-m/2}.$$

*where $c = c_m\left(\mathbb{E}_p\,|\psi(X)|^2\right)^{m/2}$ is a constant.*

**Lemma 16.** *(Large Deviations) Let $X_1, \ldots, X_n \sim p$ such that for a constant $c$, $\mathbb{E}_p |\psi(X)|^2 \leq c$ for $\psi \in \Psi_j$. Let*

$$\gamma_\psi^p = \mathbb{E}[\psi(X)],$$

$$\widehat{\gamma}_\psi = \frac{1}{n} \sum_{i=1}^{n} \psi(X_i),$$

*Let $l = \sqrt{j/n}$ and $\gamma > 0$, then, for all $j$ s.t. $2^{Dj} \in o(n)$, we have,*

$$\Pr(|\widehat{\gamma}_\psi - \gamma_\psi| > (K/2)l) \leq 2 \times 2^{-\gamma n l^2}$$

*where $K$ large enough such that*

$$\frac{K^2}{8(c + \|\psi_\epsilon\|_\infty (K/3))} > \log 2\gamma$$

Both moment and large deviation bounds from above hold for all $j$ s.t. $\mathbb{E}_p |\psi(X)|^m \leq c 2^{Dj(m/2-1)}$ which we have shown to hold for all $j \leq j_1$ (see equation 28) under both cases.

We now provide a general lemma bounding the non-linear term that we will also use when we provide a bound on the risk of the adaptive estimator.

**Lemma 17.** *Let $X_1, \ldots, X_n \sim p$ where $p$ is compactly supported and $\mathcal{F}_d = B_{p_d, q_d}^{\sigma_d}$. If $\mathbb{E}_p |\psi(X)|^{p'_d} \leq c_{p'_d} 2^{Dj(p'_d/2-1)}$, then the risk of the non-linear terms of the wavelet thresholding estimator defined above i.e.*

$$d_{\mathcal{F}} \left( \sum_{j=j_0}^{j_1} \sum_{\psi \in \Psi_j} \widetilde{\beta}_\psi \psi, \sum_{j=j_0}^{j_1} \sum_{\psi \in \Psi_j} (\beta_\psi^p + \epsilon \beta_\psi^g) \psi \right)$$

*is bounded by*

$$\sum_{j=j_0}^{j_1} 2^{-j(\sigma_d + D/2 - D/p_d)} \frac{\left\| \beta_j^p \right\|_s^{s/p'_d} + \epsilon^{s/p'_d} \left\| \beta_j^g \right\|_s^{s/p'_d}}{\sqrt{n}^{1-s/p'_d}}. \tag{32}$$

*for any $s \in [p_g, p'_d]$.*

*Proof.* We follow the procedure of Donoho et al. [1996] and Uppal et al. [2019] and break up the term into different cases. The first two of which correspond to the situation where the empirical estimate and the true value of the co-efficient are far apart. Similar to the uncontaminated case, using the large deviation bounds above we show that the probability of this happening is negligible.

This leaves us with two cases to consider: when the estimate $\widehat{\beta}_\psi$ and the true coefficient are either both small or both large. We show that both of these cases reduce to the same term which we then bound using the properties of Besov spaces and the compactness of all densities considered.

1. Let $A$ be the set of $\psi \in \Psi_j$ s.t. $\widehat{\beta}_\psi > t$ and $(1-\epsilon)\beta_\psi^p + \epsilon \beta_\psi^g < t/2$ and $r \geq 1/p'_d$ then by Hölder's inequality,

$$\sum_{j=j_0}^{j_1} 2^{-j(\sigma_d + D/2 - D/p_d)} \times \left( \sum_{\psi \in \Psi_j} \mathbb{E} |\beta_\psi^p - \widetilde{\beta}_\psi|^{p'_d} 1_A \right)^{1/p'_d}$$

$$\leq \sum_{j=j_0}^{j_1} 2^{-j(\sigma_d + D/2 - D/p_d)} \times \left( \sum_{\psi \in \Psi_j} (\mathbb{E} |\beta_\psi^p - \widetilde{\beta}_\psi|^{p'_d r})^{1/r} \Pr(A)^{1/r'} \right)^{1/p'_d}.$$

Using the large deviation and moment bound we get an upper bound,

$$\sum_{j=j_0}^{j_1} c 2^{-j(\sigma_d + D/2 - D/p_d)} \left( 2^{Dj} n^{-p'_d/2} 2^{-j\gamma/r'} \right)^{1/p'_d}$$

$$\leq c n^{-1/2} 2^{-j_0(\sigma_d - D/2 + \gamma/p'_d r')}$$

which is negligible compared to the linear term for large enough $\gamma$.

2. Let $B$ be the set of $\psi \in \Psi_j$ s.t. $\widehat{\beta}_\psi < t$ and $(1 - \epsilon)\beta_\psi^p + \epsilon\beta_\psi^g > 2t$ then same as above

$$\sum_{j=j_0}^{j_1} 2^{-j(\sigma_d + D/2 - D/p_d)} \left\| \beta_j^p + \epsilon\beta_j^g \right\|_{p_d'} 2^{-\gamma j/p_d'}$$

$$\leq 2^{-j_0(\sigma_d + \sigma_g' + \gamma)} + \epsilon \sum_{j=j_0}^{j_1} 2^{-j(\sigma_d + D/2 - D/p_d)} 2^{-\gamma j/p_d'} \left\| \beta_j^g \right\|_{p_d'}$$

which is negligible compared to the bias term and the misspecification error for large enough $\gamma$.

In other words, for the upper bounds of the first two cases we have chosen $\gamma$ (which in turn determines the value of the constant $K$ for the threshold $t = K\sqrt{j/n}$) to be large enough so that the exponent of $2^j$ in the upper bound of these two terms is negative. This enables us to upper bound the geometric series (as a sum of $j$) by a constant multiple of the first term.

3. Let $C$ be the set of $\psi \in \Psi_j$ s.t. $\widehat{\beta}_\psi > t$ and $(1-\epsilon)\beta_\psi^p + \epsilon\beta_\psi^g > t/2$ then for any $p_g \leq s \leq p_d'$,

$$\sum_{j=j_0}^{j_1} 2^{-j(\sigma_d + D/2 - D/p_d)} \times \left( \sum_{\psi \in C} \mathbb{E}\, |(1 - \epsilon)\beta_\psi^p + \epsilon\beta_\psi^g - \widetilde{\beta}_\psi|^{p_d'} \right)^{1/p_d'}$$

$$\leq \sum_{j=j_0}^{j_1} C 2^{-j(\sigma_d + D/2 - D/p_d)} \sqrt{j}^{-s/p_d'} \times \frac{\left\| (1 - \epsilon)\beta_j^p + \epsilon\beta_j^g \right\|_s^{s/p_d'}}{\sqrt{n}^{1-s/p_d'}}$$

where we have used the moment bound and the lower bound on $(1 - \epsilon)\beta_\psi^p + \epsilon\beta_\psi^g$.

4. Let $E$ be the set of $\psi \in \Psi_j$ s.t. $\widehat{\beta}_\psi < t$ and $(1-\epsilon)\beta_\psi^p + \epsilon\beta_\psi^g < 2t$ then for any $p_g \leq s \leq p_d'$:

$$\sum_{j=j_0}^{j_1} 2^{-j(\sigma_d + D/2 - D/p_d)} \left( \sum_{\psi \in E} |\beta_\psi^p|^{p_d'} \right)^{1/p_d'}$$

$$\leq \sum_{j=j_0}^{j_1} 2^{-j(\sigma_d + D/2 - D/p_d)} \times \left( \sum_{\psi \in \Psi_j} |(1 - \epsilon)\beta_\psi^p + \epsilon\beta_\psi^g|^s (2t)^{p_d'-s} \right)^{1/p_d'}$$

$$= \sum_{j=j_0}^{j_1} 2^{-j(\sigma_d + D/2 - D/p_d)} \sqrt{j}^{-s/p_d'} \times \frac{\left\| (1 - \epsilon)\beta_j^p + \epsilon\beta_j^g \right\|_s^{s/p_d'}}{\sqrt{n}^{1-s/p_d'}}$$

where we have used the upper bound on $(1 - \epsilon)\beta_\psi^p + \epsilon\beta_\psi^g$.

By applying Jensen's inequality we can show that both 3 and 4 above are bounded, for any $s \in [p_g, p_d']$, by the following (where we omit the $\sqrt{j}$ term since it only contributes a factor of polylog of $n$ or $\epsilon$ to the upper bound),

$$\sum_{j=j_0}^{j_1} 2^{-j(\sigma_d + D/2 - D/p_d)} \frac{\left\| \beta_j^p \right\|_s^{s/p_d'} + \epsilon^{s/p_d'} \left\| \beta_j^g \right\|_s^{s/p_d'}}{\sqrt{n}^{1-s/p_d'}}.$$

$\square$

We can now bound 32 by

$$\sum_{j=j_0}^{j_1} 2^{-j(\sigma_d + D/2 - D/p_d)} \frac{\left\| \beta_j^p \right\|_{p_g}^{s/p_d'} + \epsilon^{s/p_d'} \left\| \beta_j^g \right\|_{p_g}^{s/p_d'}}{\sqrt{n}^{1-s/p_d'}}.$$

Let $A$ be the set of $j$ s.t. $\left\|\beta_j^p\right\|_{p_g} \geq \epsilon \left\|\beta_j^g\right\|_{p_g}$ and $B = [j_0, j_1] \setminus A$. Then the above is upper bounded by

$$\sum_{j \in A} 2^{-j(\sigma_d + D/2 - D/p_d)} \frac{\left\|\beta_j^p\right\|_{p_g}^{p_g/p_d'}}{\sqrt{n}^{1-p_g/p_d'}} + \epsilon \sum_{j \in B} 2^{-j(\sigma_d + D/2 - D/p_d)} \left\|\beta_j^g\right\|_{p_g}$$

$$\leq \sum_{j=j_0}^{j_1} 2^{-j(\sigma_d + D/2 - D/p_d)} \frac{\left\|\beta_j^p\right\|_{p_g}^{p_g/p_d'}}{\sqrt{n}^{1-p_g/p_d'}} + \epsilon \sum_{j=j_0}^{j_1} 2^{-j(\sigma_d + D/2 - D/p_d)} \left\|\beta_j^g\right\|_{p_g}$$

$$\leq n^{1/2(p_g/p_d'-1)} 2^{-j_m((\sigma_g + D/2)p_g/p_d' + \sigma_d - D/2)} + \epsilon \sum_{j=j_0}^{j_1} 2^{-j(\sigma_d + D/2 - D/p_d)} \left\|\beta_j^g\right\|_{p_g}$$

where the second term is bounded by the misspecification error.

So for both the structured and unstructured contamination setting the bound on all terms except the misspecification error is the same. In particular, we have, for the structured setting,

$$\frac{1}{\sqrt{n}} + \frac{2^{j_0(D/2-\sigma_d)}}{\sqrt{n}} + 2^{-j_0(\sigma_d+\sigma_g-D/p_g+D/p_d')} + n^{1/2(p_g/p_d'-1)} 2^{-j_m((\sigma_g+D/2)p_g/p_d'+\sigma_d-D/2)} + \epsilon$$

which gives,

$$\frac{1}{\sqrt{n}} + n^{-\frac{\sigma_g+\sigma_d}{2\sigma_g+D}} + n^{-\frac{\sigma_g+\sigma_d-D/p_g+D/p_d'}{2\sigma_g+D-2D/p_g}} + \epsilon$$

In contrast for the unstructured setting we have,

$$\frac{1}{\sqrt{n}} + \frac{2^{j_0(D/2-\sigma_d)}}{\sqrt{n}} + 2^{-j_1(\sigma_d+\sigma_g-D/p_g+D/p_d')} + n^{1/2(p_g/p_d'-1)} 2^{-j_m((\sigma_g+D/2)p_g/p_d'+\sigma_d-D/2)}$$

$$\epsilon 2^{Dj_1(D/p_d-\sigma_d)} + \epsilon$$

At the given values of $j_0$ and $j_1$ this gives us an upper bound of

$$\frac{1}{\sqrt{n}} + \sqrt{n}^{-\frac{\sigma_g+\sigma_d}{\sigma_g+D/2}} + \sqrt{n}^{-\frac{\sigma_g+\sigma_d+D/p_d'-D/p_g}{\sigma_g+D/2-D/p_g}} +$$

$$\epsilon + \epsilon^{\frac{\sigma_g+\sigma_d+D/p_d'-D/p_g}{\sigma_g+D-D/p_g}}.$$

We note that the above proof implicitly assumes that $p_d' < \infty$ or equivalently $p_d > 1$. We provide a bound for the case $p_d' = \infty$ in the next section since in this case it is sufficient to look at linear estimators.

## B.2 Linear Rate

In this section we provide an upper bound on the risk of the linear wavelet estimator which is simply a non-linear estimator with the added constraint that $j_0 = j_1$ or without its non-linear terms. Again, in view of lemma 9 it is sufficient to consider the case $p_d' \geq p_g$.

We use the upper bounds on the components of the error of the non-linear wavelet estimator computed above with the additional constraint that $j_0 = j_1$. Therefore we have the following upper bounds along with the implied rate. In the unstructured setting, the upper bound is

$$\frac{1}{\sqrt{n}} + \frac{2^{j_0(D/2-\sigma_d)}}{\sqrt{n}} + 2^{-j_0(\sigma_d+\sigma_g-D/p_g+D/p_d')}$$

$$\epsilon + \epsilon 2^{Dj_0(D/p_d-\sigma_d)}$$

which implies the rate

$$\frac{1}{\sqrt{n}} + n^{-\frac{\sigma_g+\sigma_d+D/p_d'-D/p_g}{2\sigma_g+D-2D/p_g+2D/p_d'}} + \epsilon + \epsilon^{\frac{\sigma_g+\sigma_d+D/p_d'-D/p_g}{\sigma_g+D-D/p_g}}.$$

In the structured setting, the upper bound is,

$$\frac{1}{\sqrt{n}} + \frac{2^{j_0(D/2-\sigma_d)}}{\sqrt{n}} + 2^{-j_0(\sigma_d+\sigma_g-D/p_g+D/p'_d)} + \epsilon$$

which implies the rate

$$\frac{1}{\sqrt{n}} + n^{-\frac{\sigma_g+\sigma_d+D/p'_d-D/p_g}{2\sigma_g+D-2D/p_g+2D/p'_d}} + \epsilon.$$

For $p'_d = \infty$ the bounds on the bias or the misspecification error still hold i.e.

$$2^{-j_0(\sigma_g+\sigma_d-D/p_g)} + \epsilon + \epsilon 2^{j_0(D-\sigma_d)}$$

The variance bound is given by

$$\sum_{j=0}^{j_0} 2^{-j(\sigma_d+D/2-D/p_d)} \mathbb{E} \sup_{\psi \in \Psi_j} |\widehat{\beta}_\psi - \beta_\psi^p| \leq \frac{2^{j(D/2-\sigma_d)}}{\sqrt{n}}$$

This is shown using the lemma above bounding large deviations. We have,

$$\mathcal{P}\left( \sup_{\psi \in \Psi_j} |\widehat{\beta}_\psi - \beta_\psi^p| \geq K\sqrt{j/n} \right) \leq 2^{j(D-\gamma)}$$

which implies

$$\sum_{j=0}^{j_0} \frac{2^{j(2D-\sigma_d-\gamma)}}{\sqrt{n}} \leq \frac{1}{\sqrt{n}}$$

for $\gamma$ sufficiently large.

Now, with a choice of $2^{j_0} = n^{\frac{1}{2\sigma_g+D-2D/p_g}} \wedge \epsilon^{-\frac{1}{\sigma_g+D-D/p_g}}$ we have an upper bound of

$$\frac{1}{\sqrt{n}} + n^{-\frac{\sigma_g+\sigma_d-D/p_g}{2\sigma_g+D-2D/p_g}} + \epsilon + \epsilon^{\frac{\sigma_g+\sigma_d-D/p_g}{\sigma_g+D-D/p_g}}$$

Similarly, when contamination is structured we have a bound of

$$\frac{1}{\sqrt{n}} + n^{-\frac{\sigma_g+\sigma_d-D/p_g}{2\sigma_g+D-2D/p_g}} + \epsilon.$$

## B.3 Dense case: $p'_d \leq p_g$

Here we provide a better upper bound on the risk when $p'_d \leq p_g$ using the linear estimator from above. In this case we obtain a better bound without using the monotonicity of the dual Besov norms from lemma 9. While most of the components of the proof are the same as in the non-linear case of Section B.1 the bound on the variance is a little more involved.

*Proof.* Given $X_1, \ldots, X_n \overset{IID}{\sim} (1-\epsilon)p + \epsilon g$. Let our estimator be $\widehat{p} = \frac{1}{1-\epsilon}\widehat{p}_n$ where $\widehat{p}_n$ is the linear wavelet estimator defined above with

$$2^{j_0} = n^{\frac{1}{2\sigma_g+D}} \wedge \epsilon^{-\frac{1}{\sigma_g+D/p_d}}$$

We use the same bias variance decomposition as in the proof of the non-linear estimator with the additional constraint that $j_0 = j_1$. Therefore we have no non-linear terms to bound. We have the following upper bound on the error.

$$\mathbb{E}\, d_{\mathcal{F}_d}(\widehat{p}, p) \leq \frac{1}{1-\epsilon} \mathbb{E}\, d_{\mathcal{F}}(\widehat{p}_n, \underset{(1-\epsilon)p+\epsilon g}{\mathbb{E}} [\widehat{p}_n])$$

$$+ d_{\mathcal{F}}\left( \underset{p}{\mathbb{E}}[\widehat{p}_n], p \right) + \frac{\epsilon}{1-\epsilon} d_{\mathcal{F}}\left( \underset{g}{\mathbb{E}}[\widehat{p}_n], 0 \right)$$

where the first term is the stochastic error or the variance, the second term is the bias and the third term is the misspecification error. Here again, the bound on the bias is unchanged. Moreover, the misspecification error can be bounded in the same way as in the proof of the sparse case above. We can also show here that the variance bound remains the same but this is not as straightforward since we don't just have lower resolution terms.

Using lemma 11 we get the same bound as before on the bias or the second term above i.e.

$$d_{\mathcal{F}} \left( \mathbb{E}_p[\widehat{p}_n], p \right) \leq c2^{-j_0(\sigma_d + \sigma_g - (D/p_g - D/p_d')_+)}$$

Now we bound the variance or the first term. Let $\psi \in \Psi_j$ and

$$Y_i = \psi_(X_i) - \mathbb{E}[\psi(X)]$$

then for all $m \geq 1$, applying first the triangle inequality and then Jensen's inequality repeatedly we get

$$\begin{aligned}
\mathbb{E}[|Y_i|^m] &\leq \mathbb{E}[(|\psi(X_i)| + |\mathbb{E}[\psi(X_i)]|)^m] \\
&\leq 2^{m-1} \left( \mathbb{E}[|\psi(X_i)|^m] + |\mathbb{E}[\psi(X_i)]|^m \right) \\
&\leq 2^m \mathbb{E}[|\psi(X_i)|^m].
\end{aligned}$$

Therefore, by Rosenthal's inequality, i.e.,

**Lemma 18.** *(**Rosenthal's Inequality** (Rosenthal [1970])) Let $m \in \mathbb{R}$ and $Y_1, \ldots, Y_n$ be IID random variables with $\mathbb{E}[Y_i] = 0$, $\mathbb{E}[Y_i^2] \leq \sigma^2$. Then there is a constant $c_m$ that depends only on $m$ s.t.*

$$\mathbb{E} \left[ \left| \frac{1}{n} \sum_{i=1}^n Y_i \right|^m \right] \leq c_m \left( \frac{\sigma^m}{n^{m/2}} + \frac{\mathbb{E} |Y_1|^m}{n^{m-1}} 1_{2 < m < \infty} \right)$$

we have,

$$\begin{aligned}
&\mathbb{E}[|\widehat{\beta}_\psi - (1-\epsilon)\beta_\psi^p - \epsilon\beta_\psi^g|^{p_d'}] \\
&\leq c_{p_d'} \left( \left( \mathbb{E} |\psi(X)|^2 \right)^{p_d'/2} + \frac{\mathbb{E}[|\psi(X)|^{p_d'}]}{n^{p_d'-1}} 1_{p_d' \geq 2} \right)
\end{aligned}$$

where $c_{p_d'}$ is a constant that only depends on $p_d'$.

This implies that the variance is bounded by:

$$\sum_{j=0}^{j_0} 2^{-j(\sigma_d + D/2 - D/p_d)} \times$$

$$\left( \sum_{\psi \in \Psi_j} \mathbb{E} |\widehat{\beta}_\psi - (1-\epsilon)\beta_\psi^p - \epsilon\beta_\psi^g|^{p_d'} \right)^{1/p_d'}$$

$$\leq \sum_{j=0}^{j_0} \frac{2^{-j(\sigma_d + D/2 - D/p_d)}}{\sqrt{n}} \times$$

$$\left( \sum_{\psi \in \Psi_j} \left( \mathbb{E} |\psi(X)|^2 \right)^{p_d'/2} + \frac{\mathbb{E}[|\psi(X_i)|^{p_d'}]}{n^{p_d'/2-1}} 1_{p_d' \geq 2} \right)^{1/p_d'}$$

Now we can bound each of the terms inside the brackets separately. The second term is bounded as

$$\sum_{\psi \in \Psi_j} \frac{\mathbb{E}[|\psi(X_i)|^{p'_d}]}{n^{(p'_d/2-1)}} 1_{p'_d \geq 2}$$

$$\leq \sum_{\psi \in \Psi_j} \frac{(1-\epsilon)\mathbb{E}_p[|\psi(X_i)|^{p'_d}] + \epsilon \mathbb{E}_g[|\psi(X_i)|^{p'_d}]}{n^{(p'_d/2-1)}} 1_{p'_d \geq 2}$$

$$\leq \sum_{\psi \in \Psi_j} \frac{2^{Dj} 2^{Dj(p'_d/2-1)} + \epsilon 2^{Dj p'_d/2}}{n^{(p'_d/2-1)}} 1_{p'_d \geq 2}$$

$$\leq \sum_{\psi \in \Psi_j} \frac{2^{Dj p'_d/2}}{n^{(p'_d/2-1)}} 1_{p'_d \geq 2} \leq 2^{Dj} 1_{p'_d \geq 2}$$

While the first term is bounded as

$$\sum_{\psi \in \Psi_j} \left( \mathbb{E}|\psi(X)|^2 \right)^{p'_d/2}$$

$$\leq \sum_{\psi \in \Psi_j} \left( (1-\epsilon)\mathbb{E}_p |\psi(X)|^2 + \epsilon \mathbb{E}_g |\psi(X)|^2 \right)^{p'_d/2}$$

$$\leq \sum_{\psi \in \Psi_j} \left( (1-\epsilon) \|p\|_\infty + \epsilon 2^{Dj} w_\psi \right)^{p'_d/2}$$

where $w_\psi$ is $\int 1_{\mathrm{supp}(\psi)} g(x) dx$. Since we know that at any point at most finitely many wavelets intersect $\sum w_\psi \leq c \int g(x) dx = c$.

For $p'_d \leq 2$ by Jensen's we have,

$$2^{Dj} \left( \frac{1}{2^{Dj}} \sum_{\psi \in \Psi_j} (1-\epsilon) \|p\|_\infty + \epsilon 2^{Dj} w_\psi \right)^{p'_d/2}$$

$$\leq 2^{Dj} (c+\epsilon)^{p'_d/2} \leq 2^{Dj}$$

For $p'_d \geq 2$ again by Jensen's, we have,

$$(1-\epsilon)2^{Dj} + (\epsilon 2^{Dj})^{p'_d/2} \|w\|_{p'_d/2}^{p'_d/2}$$

$$\leq 2^{Dj} + (\epsilon 2^{Dj})^{p'_d/2}$$

where we have used the fact that $\|w\|_{p'_d/2} \leq \|w\|_1$. Since $2^{Dj_0} \leq (1/\epsilon)^{\frac{D}{\sigma_g + D/p_d}}$, for every $j \leq j_0$, $\epsilon \leq 2^{-j(\sigma_g + D/p_d)}$. This implies

$$(\epsilon 2^{Dj})^{p'_d/2} \leq 2^{j(D/p'_d - \sigma_g)p'_d/2} = 2^{j(D/2 - \sigma_g p'_d/2)} \leq 2^{Dj}$$

In conclusion, the sum of the variance terms at any resolution $j$ (not too large) is bounded by $2^{Dj/p'_d}$. Therefore, we have an upper bound for the variance term, which is the same as usual, i.e.,

$$\sum_{j=0}^{j_0} 2^{-j(\sigma_d + D/2 - D/p_d)} n^{-1/2} 2^{Dj/p'_d}$$

$$\leq \sum_{j=0}^{j_0} 2^{j(D/2 - \sigma_d)} n^{-1/2}$$

$$\leq \frac{1}{\sqrt{n}} + \frac{2^{j_0(D/2 - \sigma_d)}}{\sqrt{n}}$$

It only remains to bound the last term, or the misspecification error, which we can bound in the same as the non-linear case of section B.1 i.e. we have,

$$\frac{\epsilon}{1-\epsilon} d_{\mathcal{F}}\left(\underset{g}{\mathbb{E}}[\widehat{p}_n], 0\right) \leq c\epsilon \left(1 + 2^{j_0(D/p_d - \sigma_d)}\right)$$

Therefore, our upper bound is $\lesssim$

$$\frac{1}{\sqrt{n}} + n^{-\frac{\sigma_g + \sigma_d}{2\sigma_g + D}} + n^{-\frac{\sigma_g + \sigma_d + D/p'_d - D/p_g}{2\sigma_g - 2D/p_g + 2D/p'_d + D}} +$$

$$\epsilon + \epsilon^{\frac{\sigma_g + \sigma_d}{\sigma_g + D/p_d}} + \epsilon^{\frac{\sigma_g + \sigma_d + D/p'_d - D/p_g}{\sigma_g - D/p_g + D}}$$

$\square$

## B.4 Adaptivity

We now provide a version of the thresholding wavelet estimator above that is, under the structured contamination setting, adaptive to both the contamination proportion $\epsilon$ and smoothness of the true density $\sigma$. This essentially follows from the argument provided by Donoho et al. [1996] except that we extend it to higher dimensions. We reproduce the proof here for completeness

Given lemma 9 we only consider the case $p'_d \geq p_g$.

We now construct the adaptive version of the thresholding wavelet estimator.

Firstly, we no longer use a scaled version of $\widehat{p}_n$ but the estimator $\widehat{p}_n$ itself. This makes it adaptive to the contamination proportion $\epsilon$ and we will show that this costs us only a constant factor in the asymptotic rate. Secondly, we follow Donoho et al. [1996] and pick the following values for the resolution levels $j_0, j_1$,

$$2^{j_0} = n^{\frac{1}{D+2r}}$$

$$2^{j_1} = \left(\frac{n}{\log n}\right)^{1/D}$$

where $r$ is the regularity of the wavelets used to construct the MRA defined above. We can decompose the error as

$$\mathbb{E}\, d_{\mathcal{F}}\left(\widehat{p}_n, p\right)$$

$$\leq \mathbb{E}\, d_{\mathcal{F}}\left(\sum_{k\in\mathbb{Z}} \widehat{\alpha}_k \phi_k + \sum_{j=0}^{j_0} \sum_{\psi\in\Psi_j} \widehat{\beta}_\psi \psi, \sum_{k\in\mathbb{Z}} \alpha_k^p \phi_k^p + \sum_{j=0}^{j_0} \sum_{\psi\in\Psi_j} \beta_\psi^p \psi\right)$$

$$+ d_{\mathcal{F}}\left(\sum_{j=j_0}^{j_1} \sum_{\psi\in\Psi_j} \widetilde{\beta}_\psi \psi, \sum_{j=j_0}^{j_1} \sum_{\psi\in\Psi_j} \beta_\psi^p \psi\right)$$

$$+ d_{\mathcal{F}}\left(\sum_{k\in\mathbb{Z}} \alpha_k^p \phi_k + \sum_{j=0}^{j_1} \sum_{\psi\in\Psi_j} \beta_\psi^p \psi, p\right)$$

$$+ \epsilon d_{\mathcal{F}}\left(\sum_{k\in\mathbb{Z}} \alpha_k^g \phi_k + \sum_{j=0}^{j_1} \sum_{\psi\in\Psi_j} \beta_\psi^g \psi, 0\right)$$

$$+ \epsilon d_{\mathcal{F}}\left(\sum_{k\in\mathbb{Z}} \alpha_k^p \phi_k + \sum_{j=0}^{j_1} \sum_{\psi\in\Psi_j} \beta_\psi^p \psi, 0\right)$$

where we have an extra term at end as opposed to the non-adaptive case above. Since the density $p$ is bounded above this term is bounded by $\epsilon$ and hence by the misspecification error. The bound on the misspecification error does not change since it does not depend on the values of $j_0$ or $j_1$.

Now, since, $\sigma_g < r$ we know that the number of linear terms or $j_0$ has is smaller than above. Moreover, since $\sigma_g > D/p_g$ the number of non-linear terms $j_1$ is larger than above. Therefore, it is clear that the bias and the variance bounds hold as above. It only remains to bound the non-linear terms which from lemma 17 amounts to bounding

$$\sum_{j=j_0}^{j_1} 2^{-j(\sigma_d+D/2-D/p_d)} \frac{\left\|\beta_j^p\right\|_s^{s/p_d'} + \epsilon^{s/p_d'} \left\|\beta_j^g\right\|_s^{s/p_d'}}{\sqrt{n}^{1-s/p_d'}}. \tag{33}$$

which following the same procedure as above is bounded above by

$$\sum_{j=j_0}^{j_1} 2^{-j(\sigma_d+D/2-D/p_d)} \frac{\left\|\beta_j^p\right\|_s^{s/p_d'}}{\sqrt{n}^{1-s/p_d'}} + \epsilon \sum_{j=j_0}^{j_1} 2^{-j(\sigma_d+D/2-D/p_d)} 2^{Dj(1/p_d'-1/2)}$$

where the second term is bounded by the misspecification error. Now the first term is the same as in the case of uncontaminated setting and thereby we can bound it in the same way.

When $(2\sigma_g + D)p_g \leq (D - 2\sigma_d)p_d'$ let $s = -p_d' \frac{2\sigma_d+D-2D/p_d}{2\sigma_g+D-2D/p_g}$. Note that

$$p_g \leq -p_d' \frac{2\sigma_d + D - 2D/p_d}{2\sigma_g + D - 2D/p_g} \leq p_d'$$

where the first inequality is equivalent to the condition above and the second is equivalent to $\sigma_g \geq -\sigma_d$. We have the following bound, when we pick ,

$$\sum_{j=j_0}^{j_1} 2^{-j(\sigma_d+D/2-D/p_d)} \frac{2^{-j(\sigma_g+D/2-D/p_g)s/p_d'}}{\sqrt{n}^{1+\frac{2\sigma_d+D-2D/p_d}{2\sigma_g+D-2D/p_g}}}$$

$$\asymp n^{-\frac{\sigma_g+\sigma_d+D/p_d'-D/p_g}{2\sigma_g+D-2D/p_g}}$$

as desired (where we omit any $\log(n)$ terms).

Now, when $(2\sigma_g + D)p_g \geq (D - 2\sigma_d)p_d'$ the error of the non-linear terms is bounded by the error of the first non-linear term i.e. $j = j_0$. We can bound the error of the non-linear terms for all $j \geq j_0^*$ where $j_0^*$ is the original non-adaptive threshold i.e. $2^{j_0} = n^{\frac{1}{2\sigma_g+D}}$. For the extra terms between $j_0$ and $j_0^*$ we show that in this range the error of the non-linear terms cannot be worse that the linear error. The large deviation terms (1) and (2) above are negligible by the same argument as above. The terms (3) and (4) can be trivially bounded by

$$2^{-j(\sigma_d+D/2-D/p_d)} \frac{2^{Dj/p_d}}{\sqrt{n}} = \frac{1}{\sqrt{n}} 2^{j(D/2-\sigma_d)}$$

which is bounded by the linear rate.

## C  Lower Bounds

In this section we prove our lower bounds. We first provide lower bounds in the case of structured contamination since these also hold for the case of unstructured contamination. We then provide additional lower bounds that are specific to the unstructured case.

### C.1  Structured Contamination

We assume here that $G$ has a density that lives in a Besov space i.e. $\mathcal{F}_c = B_{p_c,q_c}^{\sigma_c}$.

*Proof.* We will use Fano's lemma to imply lower bounds here.

First we show that the lower bounds on the risk in the setting of no contamination also bound the risk in the contaminated setting. The key idea here is that if the set of densities chosen to provide bounds in the uncontaminated setting (when $\epsilon = 0$) are perturbations of a "nice" density $p_0$, then in

the contaminated setting we can choose our contamination density to be this nice density $g_0$. This will imply that the contamination does not affect the samples i.e. the samples are generated merely from the perturbation (since $(1-\epsilon)(g_0 + p_\tau) + \epsilon g_0 = g_0 + (1-\epsilon)p_\tau$ where $p_\tau$ is some perturbation).

We first state Fano's lemma.

**Lemma 19.** *(Fano's Lemma; Simplified Form of Theorem 2.5 of Tsybakov [2009]) Fix a family $\mathcal{P}$ of distributions over a sample space $\mathcal{X}$ and fix a pseudo-metric $\rho : \mathcal{P} \times \mathcal{P} \to [0, \infty]$ over $\mathcal{P}$. Suppose there exists a set $T \subseteq \mathcal{P}$ such that there is a $p_0 \in T$ with $p \ll p_0 \; \forall p \in T$ and*

$$s := \inf_{p,p' \in T} \rho(p, p') > 0 \quad , \quad \sup_{p \in T} D_{KL}(p, p_0) \leq \frac{\log |T|}{16},$$

*where $D_{KL} : \mathcal{P} \times \mathcal{P} \to [0, \infty]$ denotes Kullback-Leibler divergence. Then,*

$$\inf_{\widehat{p}} \sup_{p \in \mathcal{P}} \mathbb{E}\left[\rho(p, \widehat{p})\right] \geq \frac{s}{16}$$

*where the* inf *is taken over all estimators $\widehat{p}$.*

Now we choose our set of densities as,

$$p = p_0 \qquad\qquad\qquad p_\tau = p_0 + \frac{1}{(1-\epsilon)}c_g f_\tau$$

$$g = p_0 \qquad\qquad\qquad g_\tau = p_0.$$

where $p_0 + c_g f_\tau \in \mathcal{F}_g$ for every $\tau$. Notice that the KL divergence remains unchanged from the uncontaminated setting,

$$KL((1-\epsilon)p_0 + \epsilon p_0, (1-\epsilon)(p_0 + \frac{1}{1-\epsilon}c_g g_\tau) + \epsilon p_0)$$
$$= KL(p_0, p_0 + c_g f_\tau)$$

i.e. the KL divergence doesn't depend on the existence of contamination. Neither does $d_{\mathcal{F}_d}(p_\tau, p_{\tau'})$. Since, $1 - \epsilon \in [1/2, 1]$ we can treat it as a constant and only write $c_g$ henceforth. Therefore we are essentially in the case of no contamination i.e. if there exist densities $p, p_\tau$ indexed by $\tau$ such that they satisfy the assumptions of Fano's lemma then the conditions of Fano's lemma are also satisfied for $(1-\epsilon)p + \epsilon g$, $(1-\epsilon)p_\tau + \epsilon g_\tau$. Moreover, the distance we want to bound i.e. $d_{\mathcal{F}_d}(p_\tau, p_{\tau'})$ does not depend on the contamination either. Therefore, we have a lower bound here that is the same as the one in the setting with no contamination.

We note that the densities used to prove the lower bound in Uppal et al. [2019] are exactly of this form (see section B of the appendix) (Uppal et al. [2019] study the uncontaminated version of this problem). Therefore, their lower bound (see Theorem 4) is implied here i.e.

$$C\left(\frac{1}{\sqrt{n}} + n^{-\frac{\sigma_g + \sigma_d}{2\sigma_g + D}} + \left(\frac{\log n}{n}\right)^{\frac{\sigma_g + \sigma_d - D/p_g + D/p'_d}{2\sigma_g + D - 2D/p_g}}\right)$$

Second, we consider the case where we "move" the perturbation so that the samples are generated from the same density. In particular, we first perturb the contamination and then we move this perturbation to the true density i.e.

$$p = p_0 \qquad\qquad\qquad \widetilde{p} = p_0 + \frac{\epsilon}{1-\epsilon}c\psi_\epsilon$$

$$g = g_0 + c\psi_\epsilon \qquad\qquad\qquad \widetilde{g} = g_0$$

Then the KL divergence between the densities that generate the samples is zero since they are the same ($(1-\epsilon)p + \epsilon g$ is the same in both cases). It is easy to see that $\widetilde{p}, g$ both live in the respective density classes i.e. $\mathcal{F}_g, \mathcal{F}_c$ for a small enough constant $c$. Using Le Cam's two point argument i.e.

**Lemma 20.** *(Le Cam (see section 2.3 of Tsybakov [2009])) Let $P_1$, $P_2$ be two probability measures on $\mathcal{X}$ s.t. $d(P_1, P_2) = s$. If $KL(P_1, P_2) \leq \alpha < \infty$ then, for any $\widehat{P}$*

$$\mathbb{E}_{P_i}[d(\widehat{P}, P_i)] \geq \frac{s}{8}e^{-\alpha}$$

we have a lower bound that is the distance between $p_0$ and $p_0 + \frac{\epsilon}{1-\epsilon}\psi_\epsilon$ i.e.

$$d_{\mathcal{F}_d}(p_0, p_0 + \frac{\epsilon}{1-\epsilon}\psi_\epsilon) = \epsilon.$$

$\square$

This section provided lower bounds on the risk that are minimax in the structured contamination setting. We now provide additional bounds that hold when we have no structural assumptions on the contamination.

## C.2 Unstructured Contamination

Here we assume only that $g$ is a compactly supported probability density. We will pick a single perturbation of a "nice" density and use this to construct the contamination densities in such a way that the data is generated from the same density. Hence, the KL divergence between the data generating densities will be zero. Then, as before, we can apply Le Cam's two point argument to bound the risk.

### C.2.1 Sparse or Lower Smoothness Case

Let $p = g_0, \widetilde{p} = g_0 + c_g\psi_0$ for some $\psi_0 \in \Psi_j$. Now we can pick densities $g, \widetilde{g}$ such that

$$(1 - \epsilon)p + \epsilon g = (1 - \epsilon)\widetilde{p} + \epsilon\widetilde{g}$$

if and only if

$$g - \widetilde{g} = \frac{(1 - \epsilon)}{\epsilon}(\widetilde{p} - p)$$

integrates to zero and its $L_1$ norm is $\leq 2$ (see Lemma 6.6 of Liu and Gao). For $\widetilde{p}$ to be a density in $\mathcal{F}_g$ we need

$$c_g \leq c \min(2^{-Dj/2}, 2^{-j(\sigma_g + D/2 - D/p_g)})$$

Since $\sigma_g \geq D/p_g$, we let $c_g = 2^{-j(\sigma_g + D/2 - D/p_g)}$. From the above constraint on the $L_1$ of $g - \widetilde{g}$ norm we need

$$\frac{c_g}{\epsilon} 2^{-Dj/2} \|\psi\|_\infty \leq 2$$

This is equivalent to

$$2^{-j(\sigma_g + D - D/p_g)} \leq c\epsilon$$

where $c$ is a constant. We pick $2^j = \epsilon^{-\frac{1}{\sigma_g + D - D/p_g}}$. We also choose a simple discriminator i.e.

$$\Omega_d = \{c_d\psi_0\}$$

where $c_d = 2^{-j(\sigma_d + D/2 - D/p_d)}$ so that $\Omega_d \subseteq \mathcal{F}_d$. Then, by 20 the minimax risk is lower bounded by

$$
\begin{aligned}
d_{\mathcal{F}_d}(p, \widetilde{p}) &\geq d_{\Omega_d}(p, \widetilde{p}) \\
&\gtrsim c_g c_d \\
&= c2^{-j(\sigma_g + \sigma_d + D - D/p_g - D/p_d)} \\
&= \epsilon^{\frac{\sigma_g + \sigma_d + D/p'_d - D/p_g}{\sigma_g + D - D/p_g}}.
\end{aligned}
$$