[Reviews · NeurIPS 2020]

Review 1

Summary and Contributions: The author(s) show the robustness of nonparametric density estimations under Besov integral probability metrics for mixture distributions with an unknown outlier distribution. Therefore, they prove the minimax convergence of a wavelet estimator.

Strengths: The authors provide a theoretical sound framework showing the convergence of wavelet estimator for (robust) density estimation for different contaminations (structured and unstructured). As this is relevant for many applications and its relatedness to GANs is shown, I consider it interesting. Further, in most cases, it is well written and references to prior work used for different conclusions are also present.

Weaknesses: Some (minor) spelling mistakes such as “theorem 5” (e.g., line 183 and 297) or “of the form 7” (295, 306).Similarly, more detailed references to the appendix would be preferable. More details regarding the implications for real-world data would be beneficial. A test on synthetic data for different contaminations would be ideal. However, as the article is theoretical, this is not necessarily required.

Correctness: Yes, the paper seems to be technically sound.

Clarity: Yes, the paper is well written.

Relation to Prior Work: It is clearly discussed how their findings relate to recent work such as Uppal et al. [2019], Chen et al. [2016], and Liu and Gao [2017]. They also include a discussion on how their work relates to the earlier work of Kim and Scott [2012] and Vandermeulen and Scott [2013].

Reproducibility: Yes

Additional Feedback: Figure 1 requires some more explanation in the caption. I would prefer consistent writing of form and equation. Also, many equations do not have numbers. It is not clear where the claim in that a GAN with “ReLu activations can learn the distribution of the ...” is shown. This statement needs clarification. I would prefer citations of the final work over arxiv, e.g., Uppal et al. [2019] vs. the publication at NIPS. Line 305 : “estimator of 4.2”, what is 4.2 referring too? In contrast to other sections, section 5 is harder to understand in contrast to the other sections.


Review 2

Summary and Contributions: The paper discusses minimax risks for non-parametric density estimation under a large and common family of losses (integral probability metrics) with contaminated data. The authors prove minimax rates using wavelet thresholding estimators shown to be optimal.

Strengths: - The paper is clearly relevant to the Neurips community, studying robustness of estimation under unknown contamination of the available data. - In comparison with related works, the paper brings novel contributions in two directions: (i) by finding optimal density estimators (with or without contamination) which only depend on the data and don’t require any knowledge on the underlying distribution. (ii) extending the computation of minimax rates under contamination to wider classes of losses and densities of contaminations. - Furthermore, implications of the presented results for ideal GANs are discussed.

Weaknesses: - Under my relative level of expertise, I do not find serious weaknesses of the paper.

Correctness: - Given my relative level of expertise and time allowed, I did not check the proofs of the paper.

Clarity: -The paper is clear, including intuitions guiding the reader where possible.

Relation to Prior Work: - The relationship to prior work is clearly discussed in a dedicated section.

Reproducibility: Yes

Additional Feedback: Typo / form: - Maybe it would be preferable to have numbering of all equations. - line 102, could the authors check the definition of the Haar wavelets scaled from the mother wavelet? There is no \lambda in the definition. - line 193 “the the” ******* I have read other reviews and authors feedback, it confirms my positive assessment of the paper.


Review 3

Summary and Contributions: This paper studies convergence of minimax estimators for a large class of losses commonly used in machine learning (Besov integral probability metrics) for a model of polluted data. They describe a data-dependent estimator (i.e., no information about the smoothness of the space is required) which they prove yields minimax optimal convergence rates. Finally, they describe the implications for GAN convergence.

Strengths: This paper has very strong theoretical results, several of which are genuinely surprising. First, they derive a minimax rate for the "unstructured" Huber contamination model in which there is no smoothness restriction on the contaminating distribution (though it must be compactly supported). This rate, interestingly, is better than the rate achieved for density estimation at a point, a very interesting result that certainly merits more discussion in the paper---it would be nice to have a sense of why this is the case. The rates reveal some interesting conclusions, as well. In particular, the observation that asymptotically, it is primarily the boundedness of the contaminating distribution and that further smoothness assumptions do not improve the rate. This is a really nice result.

Weaknesses: Added after feedback: I do think that some numerical experiments on synthetic data could be illuminating, as indicated by the other reviewers. I maintain my rating of this very strong paper, but encourage the authors to consider designing experiments to probe the convergence rates. The authors could significantly improve the main-text discussions of the proofs. There is essentially no information about proof technique, whether or not the steps are standard, etc. Incorporating some description of this type would certainly make the paper more readable. Another issue I encountered was that I felt the description of the wavelet estimator (in particular how it differed from that of Donoho) was vague and not really addressed in the main text.

Correctness: Yes, the claims seem correct and the assumptions reasonable.

Clarity: This is a very well-presented paper. Clear, economical language and well-structured.

Relation to Prior Work: Yes, it's very well-contextualized within a somewhat narrow literature.

Reproducibility: Yes

Additional Feedback:


Review 4

Summary and Contributions: This paper considers a wide range of smoothness conditions lying in a Besov space in the family of loss function called integral probability metric. The paper then details the convergence rates for a purely data-dependent density estimators and non-parametric convergences guarantees for data which have been contaminated by random outliers.

Strengths: The paper reads very well. The authors first introduce a formal problem statement of the contamination density for both the unstructured and structured setting. Then definitions of Besov space and linear estimator is given. The authors provide a discussion of the related work well to show the contributions of this work. The related work section discusses the limitations and assumptions that have been made in current approaches found in literature. The authors provide theorems to show the minimax rate for both unstructured and structured contamination.

Weaknesses: I like the idea that the authors have given examples and practical implications on applying this approach to GANs. It would be great if the authors give more examples or an empirical evaluation to demonstrate the capability. This will greatly improve the understanding of the theoretical contributions outlined in this paper.

Correctness: I do not have the technical expertise to comment in detail for this section. I have read the main submission and majority of the content appears to be correct to me.

Clarity: Overall the paper is well structured and written. The structure of the paper is a bit unconventional by first providing the technical backgrounds before highlighting the related work. But, I quite like this because it will allow the authors to use mathematical notations to explain the related work section. The part which I have found the most difficult to follow is Section 2.1, where I am unfamiliar with a lot of the mathematical notations. For the short timeframe I have to review this paper, I was unable to understand the meaning behind each of the mathematical notation. However, that being said paper is out-of-area for me. So, I am unable to appropriately assess whether section is well written or not.

Relation to Prior Work: The authors have provided a detail discussion of the prior work in both the introduction and Section 3 Related Work. This work is different from the previous contributions by providing convergence rates for a purely data-dependent density estimation for Besov IPMs.

Reproducibility: Yes

Additional Feedback: Overall, the paper addresses a very important issue into density estimation when contaminated by random outliers which is encountered in many machine learning problems. The theoretical guarantees for the linear and non-linear convergences rate and minmax bound is extremely useful for designing robust machine learning models. Unfortunately, I do not have the technical expertise to comment on the correctness of this approach. But for an out-of-area reviewer, I can see that this paper is well motivated and is written and structured well. ------------------------------------------- Post-Author Feedback Comments: ------------------------------------------- Thank you for your response. As mentioned in my original review, experimental results with synthetic data could strengthen the paper and improve understanding of the paper, but it is not critical. The authors have explained using examples for potential applications for the theoretical results in Section 4.3 which seems good enough for me. Also, looking back at my original rating of 6 does seem a bit harsh for the minor criticism I have given. I have now increased it to 7.

[Author Response · NeurIPS 2020]

1 We thank all the reviewers for their comments and suggestions.

2 **Reviewers #1 & #4:**

3 **"More details regarding the implications for real-world data would be beneficial."**
4 Due to space constraints we omitted a discussion about the "breakdown point" which may have practical implications.
5 The "asymptotic breakdown point" is the maximum asymptotic proportion $\epsilon$ of outlier samples such that the estimator
6 still converges at the uncontaminated optimal rate; i.e., the proportion of corruption the estimator can tolerate before
7 performance degrades.

Figure 1: Asymptotic breakdown point when $p'_d = \infty$ and $\sigma'_g = \sigma_g - D/p_g$.

Figure 1 illustrates the asymptotic breakdown point for the case $p'_d = \infty$ as a function of $\sigma_d$; this includes as special cases the $L_\infty$ loss and Kolmogorov-Smirnov metric.

14 In Figure 2 we pick $\sigma'_g = D = 5$ for the above setting to show how the breakdown point increases as a function of $\sigma_d$
15 (smoothness of functions that define the IPM; larger smoothness implies weaker loss). So, estimation under a weaker
16 loss may be more robust to contamination. We will add more discussion about how understanding breakdown points
17 could be useful in the context of real-world data.

18 **"A test on synthetic data for different contaminations would be ideal."**
19 We felt it would be difficult for experiments on synthetic data to elucidate our results,
20 due to their asymptotic nature. However, if the reviewers feel this would help clarify
21 the significance of the paper we will add it for the camera ready version.

22 **"It is not clear where the claim that a GAN with 'ReLu activations can learn the**
23 **distribution of the ...' is shown."**
24 In corollary 6 we extend the result of Uppal et. al. [2019] to show that a perfectly
25 optimized GAN estimate (of the form of eq. (7)) with large enough fully-connected
26 ReLU generator and discriminator networks converges (under Besov IPMs) to the true
27 Besov distribution at the minimax optimal rate, both adaptively and in the presence of
28 contamination. We will clarify this in the paper.

Figure 2: Exponent of $n$ in breakdown point as a function of $\sigma_d$.

29 **"Line 305 : "estimator of 4.2", what is 4.2 referring to?"**
30 We meant to refer to the wavelet thresholding estimator defined in section 4.

31 **Reviewer #2:**

32 **"line 102, could the authors check the definition of the Haar wavelets scaled from the mother wavelet? There is**
33 **no $\lambda$ in the definition."**
34 In dimension $D$ there are $2^D - 1$ mother wavelets $\psi_\epsilon$ indexed by $\epsilon \in \{0,1\}^D \setminus (0,\ldots,0)$. For each of these mother
35 wavelets, daughter wavelets $2^{Dj/2}\psi_\epsilon(2^{Dj}x - k)$ are constructed by translation (by integers $k$) and scaling (by $2^{-Dj}$
36 horizontally and $2^{Dj/2}$ vertically). At resolution $j$, such wavelets can be indexed by $\lambda = 2^{-j}k + 2^{-j-1}\epsilon$, as any such
37 value of $\lambda$ uniquely defines $k$ and $\epsilon$. For example, in dimension $D = 1$, $\lambda = 1.5$ implies that $j = 0$ and $k = 1$. We will
38 clarify this re-indexing in the paper.

39 **Reviewer #3:**
40 **"This rate, interestingly, is better than the rate achieved for density estimation at a point"**
41 The rates differ primarily in the $\epsilon$ term i.e. $\epsilon^{-\frac{\sigma_0}{2\sigma_0+1}}$ vs $\epsilon^{-\frac{\sigma_0}{2\sigma_0+1/2}}$ (section 3.2) which is obtained when the misspecifica-
42 tion error dominates the variance. The misspecification error $\epsilon d_{\mathcal{F}_d}(\mathbb{E}_g[\widehat{p}_n])$ (eq. (19)) is the error due to contamination.
43 The point-wise misspecification error scales as the $\mathcal{L}^\infty$ error or $\epsilon 2^{Dj}$ with resolution $j$ which is larger than the $\mathcal{L}^{p'_d}$ error
44 which scales as $\epsilon 2^{Dj/p_d}$. Intuitively, this reflects the fact that while under both losses the "worst-case" contamination
45 densities are "spikes" concentrated around single points (rather than, say, uniform over the domain), the $\mathcal{L}^\infty$ loss, as
46 well as estimation at a point, are more sensitive to such contamination than, e.g., $\mathcal{L}^2$ loss. We will try to clarify this in
47 the paper.

48 **"the description of the wavelet estimator (in particular how it differed from that of Donoho)"**
49 The wavelet estimator that we have used is similar to that of Donoho et. al. [1996]. We merely chose to estimate the linear
50 terms (up to resolution $j_0$) with basis $(\cup_k (\phi_{0k} \cup \bigcup_{0 \le j \le j_0} \psi_{jk}))$ as opposed to $\cup_k \phi_{jk}$. These bases are equivalent.
51 However, no prior work has studied the *robustness properties of this estimator in the presence of contamination*. As the
52 reviewer suggested, we will add discussion of the novel aspects of the proofs to the main paper.

[Meta-Review · NeurIPS 2020]

The reviewers agree that this paper would make a worthy contribution to NeurIPS. Please see the reviews for ways to improve the paper (especially regarding clarity and real world data). Experimental results with synthetic data could strengthen the paper but are not critical, if you think they could improve understanding of the paper, you might want to include it in the supplementary material.